# Pan-tissue transcriptome analysis reveals sex-dimorphic human aging

Siqi Wang[1,2]*, Danyue Dong[1], Xin Li[1], Zefeng Wang[1,3]*

[1]Shanghai Institute of Nutrition and Health, University of Chinese Academy of Sciences, Chinese Academy of Sciences, Shanghai, China; [2]University of California, Los Angeles, Los Angeles, United States; [3]School of Life Science, Southern University of Science and Technology, Shenzhen, China

## eLife Assessment

In this study Wang et. al. mined publicly available RNA-seq data from The Genotype-Tissue Expression (GTEx) database spanning multiple tissues to ask the question of how transcriptomes are changed with age and in both sexes. The authors provide **solid** evidence reporting widespread gene expression changes and alternative splicing events that vary in an age- and sex-dependent manner. An **important** finding is that many of these changes coincide with the time sex hormones begin to decline; additionally, the rate of aging is faster in males than in females.

*For correspondence:
siqiwang@ucla.edu (SW);
wangzf@sustech.edu.cn (ZW)

Competing interest: The authors declare that no competing interests exist.

**Abstract** Complex diseases often exhibit sex dimorphism in morbidity and prognosis, many of which are age-related. However, the underlying mechanisms of sex-dimorphic aging remain foggy, with limited studies across multiple tissues. We systematically analyzed ~17,000 transcriptomes from 35 human tissues to quantitatively evaluate the individual and combined contributions of sex and age to transcriptomic variations. We discovered extensive sex dimorphisms during aging with distinct patterns of change in gene expression and alternative splicing (AS). Intriguingly, the male-biased age-associated AS events have a stronger association with Alzheimer's disease, and the female-biased events are often regulated by several sex-biased splicing factors that may be controlled by estrogen receptors. Breakpoint analysis showed that sex-dimorphic aging rates are significantly associated with decline of sex hormones, with males having a larger and earlier transcriptome change. Collectively, this study uncovered an essential role of sex during aging at the molecular and multi-tissue levels, providing insight into sex-dimorphic regulatory patterns.

## Introduction

Human diseases often exhibit differences between females and males, including sex-differential morbidity, prognosis, and mortality (*Westergaard et al., 2019*; *Khramtsova et al., 2019*; *Regitz-Zagrosek, 2012*). The sex dimorphism has been widely reported in neurological disorders (*Ferretti et al., 2018*; *Jazin and Cahill, 2010*), cardiovascular diseases (*Colafella and Denton, 2018*), immunological defects (*Klein and Flanagan, 2016*), and cancers (*Li et al., 2018b*). Accordingly, the life expectancy shows substantial variability between females and males. The evidence on sex-differential mortality was recently reported during the COVID-19 pandemic, mainly affected by different immune responses between females and males (*Park, 2020*; *Takahashi et al., 2020*). The known molecular mechanisms mostly revolved around sex-differential genetic variants, epigenetics, transcriptomes, and sex-differential responses to environmental exposures (*Khramtsova et al., 2019*). It is worth noting that many diseases with sex dimorphism are also age-related, especially in neurodegenerative disorders, cardiovascular diseases, and cancers (*Sampathkumar et al., 2020*). Such sex-differential

susceptibility contributes substantially to different life expectancies between females and males, and the related underlying mechanisms are thought to be hormone-driven, mitochondria-related, and sex-chromosome-linked (*Sampathkumar et al., 2020*; *Hägg and Jylhävä, 2021*; *Austad and Fischer, 2016*; *Davis et al., 2021*; *Ferretti and Santuccione Chadha, 2021*). Previously, sex-differential aging signatures, including chronological trends and gene networks, have been studied in the whole blood and brain regions of healthy donors (*Márquez et al., 2020*), revealing the sex differences of disease vulnerabilities during aging (*Berchtold et al., 2008*; *Zhou et al., 2023*). Sex differences regarding the composition and inflammatory signaling of immunocytes in blood were also analyzed in single-cell resolution, which suggested sex-differential aging in the immune response (*Huang et al., 2021*). Moreover, at the proteogenomic level, sex-biased genes play key roles in several important cellular processes during cardiac aging, such as mitochondrial metabolism, RNA splicing, and translation, implying sex dimorphism in cardiac diseases (*Han et al., 2022*). However, a systematic analysis across multiple tissues on the sex-differential aging and underlying molecular mechanisms is currently lacking.

The Genotype-Tissue Expression (GTEx) project contains a large set of high-throughput sequencing data from postmortem donors across 54 human tissues (*Aguet et al., 2020*; *Oliva et al., 2020*). Previous studies have uncovered many sex-biased genes or expression quantitative trait loci (eQTLs) (*Oliva et al., 2020*; *Khodursky et al., 2022*) across different tissues that could affect several critical cellular signal pathways related to human diseases. Since the GTEx donors have a wide age range of 20–70 years, it is possible to identify genes and molecular pathways significantly changed during the aging process (*Zeng et al., 2020*; *Jia et al., 2018*). However, the relationship between sex-differential regulation and aging has not been thoroughly studied on a transcriptome or pan-tissue level scale. Therefore, the GTEx data could also provide an integrated resource for a comprehensive study of this topic.

In addition to transcriptional regulation, most human genes undergo alternative splicing (AS) to produce different isoforms with distinct activities (*Gallego-Paez et al., 2017*). AS is tightly regulated in various tissues or developmental stages by multiple *cis*-acting elements and *trans*-acting factors (*Wang and Burge, 2008*; *Baralle and Giudice, 2017*). Dysregulation of AS is closely related to many age-related diseases, such as neurodegenerative diseases (*Nikom and Zheng, 2023*) and cancers (*Wang et al., 2020*). Since most introns are co-transcriptionally spliced, AS is also affected by various factors that regulate transcription, such as promoter activity, Pol II elongation, and chromatin modification/remodeling (*Baralle and Giudice, 2017*). Recent studies reported that AS regulation is closely associated with sex (*Karlebach et al., 2020*) and age (*Wang et al., 2018*; *Bhadra et al., 2020*; *Angarola and Anczuków, 2021*) thus, it is intriguing to further explore the AS regulation during aging stages in different sexes.

In this study, we initiated an integrated analysis of pan-tissue transcriptome data in GTEx and systematically determined how sex and age contributed to the global variations of gene expression (GE) and AS (*Figure 1*). We further focused on the sex-biased AS changes during aging and found that these AS events are significantly associated with neurodegenerative diseases in a sex-biased fashion. We further examined the chronological changes of age-associated genes and found a sex-dimorphic transcriptome aging, with males showing an earlier onset of aging and a faster aging rate in most tissues. Collectively, this study clarified the sex and age effects on the transcriptomic changes, revealing the sex-dimorphic aging at a multi-tissue level and their potential mechanisms.

## Results

### Sex and age are critical drivers to the global transcriptome variation

To systematically study how sex and age affect transcriptome complexity, we conducted a thorough analysis of RNA-seq data from the GTEx project (*Aguet et al., 2020*) to quantify the GE and AS patterns of all genes from 54 tissues (*Figure 1*). Due to data sparseness, the brain tissues were recombined into four functional regions (*Supplementary file 1*), including hormone- or emotion-related region, movement-related region, memory-related region, and decision-related region (see Methods). We performed the principal component analysis (PCA) on GE and AS data and observed the sex and age differences in several tissues (*Figure 1—figure supplement 1*). Based on a previous study (*Lopes-Ramos et al., 2020*), we designed a method called principal component-based signal-to-variation ratio (pcSVR) to measure the adjusted Euclidean distance in higher dimensions. The pcSVR quantified

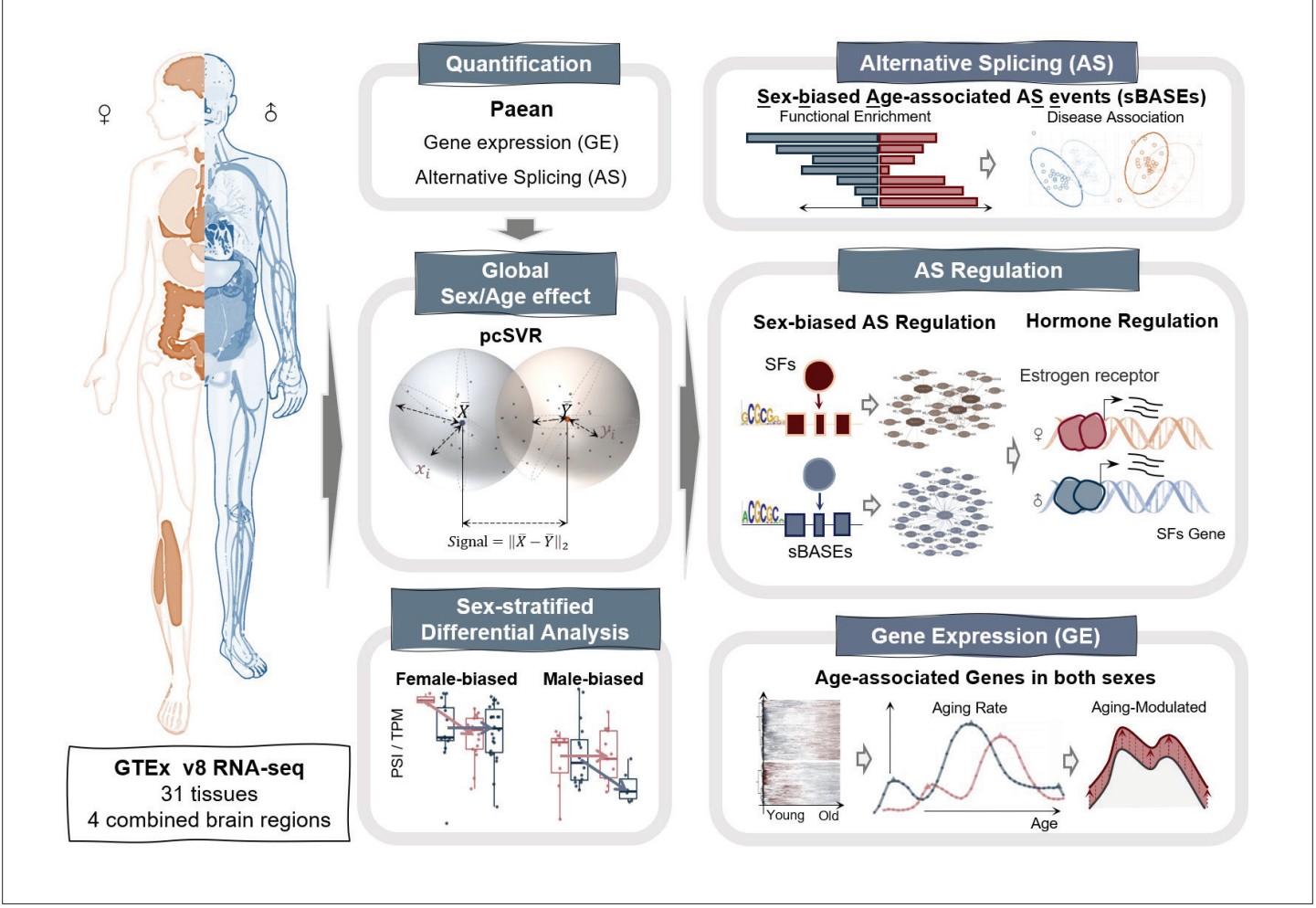

**Figure 1.** Schematic overview of our study. RNA-seq datasets from 54 human tissues (including 13 brain regions) were downloaded from GTEx Portal. Gene expression (GE) and alternative splicing (AS) quantifications were performed by Paean. The individual sex or age and age-by-sex effects on global transcriptomic variation were evaluated by designing a principal component-based signal-to-variation ratio (pcSVR) value, and the age-differential genes/AS events were subsequently identified in each sex. Sex-biased age-associated AS events were used to find the sex-specific associations with certain diseases. We also constructed AS regulatory networks to find the regulatory age-associated splicing factors (SFs) and explored the transcriptional regulation of sex hormones via nuclear receptors. Moreover, we focused on age-associated GE and then evaluated the aging rate and their modulated genes in different sexes.

The online version of this article includes the following figure supplement(s) for figure 1:

**Figure supplement 1.** Principal component analysis (PCA) on gene expression (GE) and alternative splicing (AS) profiles.

the distance between different sex or age groups divided by the data dispersion within each group (intrinsic differences between groups will result in a pcSVR value significantly larger than 1), serving as a reliable measurement for the sex or age effects on transcriptomic variations (*Figure 2—figure supplement 1A*, see Materials and methods, *Equations 1–3*). Compared with the direct identification of differentially expressed genes or AS events, pcSVR provides a global measurement by considering variations from all genes and AS events between different groups.

We grouped the samples into male vs. female and young vs. old and calculated the sex-pcSVR and age-pcSVR in multiple tissues (*Figure 2A, B*). Due to a large variation of menopause ages among different individuals and a continuous decline of sexual steroid levels between ages 40 and 60 (*Chan et al., 2020*; *Ober et al., 2008*), we grouped the samples into young (age <40) vs. old (age >60) with an age gap instead of a specific age cutoff to reduce the data noise. We found that the breast tissue showed the largest sex-pcSVR, while the adipose and cardiovascular tissues showed larger age-pcSVR (i.e., differences between age groups). The tissues with significant sex differences were similar

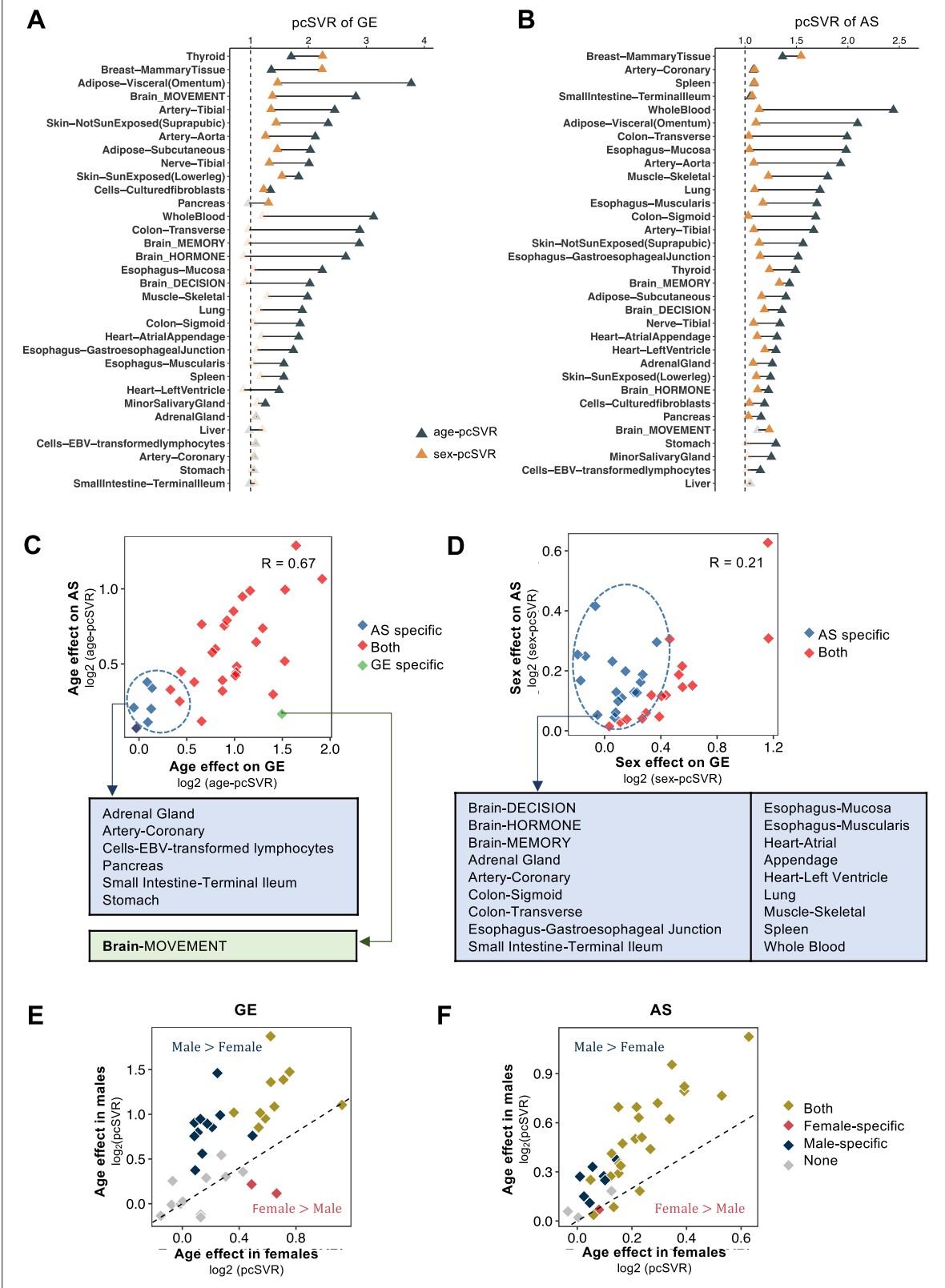

**Figure 2.** Individual sex or age effects and combined age-by-sex effects on global transcriptome variation. Principal component-based signal-to-variation ratio (pcSVR) between different sex or age groups calculated by gene expression (GE) (**A**) and alternative splicing (AS) (**B**). The yellow triangles show the significant sex-pcSVR calculated between females and males, while black triangles show the significant age-pcSVR calculated between young (i.e., 20–40 years old) and old (i.e.,>60 years old) using the empirical p-value cutoff 0.1. Insignificant data points are labeled in gray. (**C**) The age effect

*Figure 2 continued on next page*

*Figure 2 continued*

on transcriptome variation as calculated by GE vs. AS. The *X*-axis of the scatter plot shows the log$_2$ transformed age-pcSVR between young and old calculated by GE, while the *Y*-axis shows the values calculated by AS. Blue dots indicate the tissues with significant pcSVR specific in AS, while green dots indicate the tissues with significant pcSVR specific in GE. Tissue names in each class are labeled under the scatter plot. (**D**) The sex effect on transcriptome variation as calculated by GE vs. AS. The analyses and labels are similar to panel **C** except the sex-pcSVR is used in all calculations. Scatter plots of the age effect between females *vs.* males calculated by GE (**E**) and AS (**F**). The *X*-axis shows log$_2$ transformed age-pcSVR calculated in females, while the *Y*-axis shows the age effect calculated in males. Dashed lines showed equal age-pcSVR in females and males. Dark red and blue dots indicate the tissues with female- and male-specific age effects.

The online version of this article includes the following figure supplement(s) for figure 2:

**Figure supplement 1.** Principal component-based signal-to-variation ratio (pcSVR) calculation and robustness.

to those reported in a previous study of sex-differential GE using transcriptomic signal-to-noise ratio (SNR) (*Lopes-Ramos et al., 2020*), including breast, thyroid, skin, and adipose. Using the permutation test for pcSVRs, we found that age showed substantially larger effects than sex to human transcriptome in most tissues as judged by both GE and AS (*Figure 2A, B*, *Supplementary file 2*). Moreover, AS was significantly affected by both sex and age across most tissues, while GE was affected by sex in a much smaller number of tissues as compared to AS profiles (i.e., comparing the orange triangles in *Figure 2A vs. B*). For example, the coronary artery and adrenal gland are significantly affected by sex and age in their AS profile, but their GE profiles are not affected by sex or age (*Figure 2A, B*). Similar results were found even when removing the genes and AS events encoded by sex chromosomes (*Figure 2—figure supplement 1B, C*), selecting a range of PC cutoffs to capture different global variance (*Figure 2—figure supplement 1D, E*), or using other approaches for p-value estimation (e.g., bootstrapping with replacement, see Materials and methods, *Figure 2—figure supplement 1F, G*).

We further compared the age and sex effects calculated by GE vs. AS (*Supplementary file 2*) and found a significant correlation between the age-pcSVR of GE and AS in most tissues ($R = 0.67$, p < 0.05, *Figure 2C*), suggesting the age contribution to transcriptome variation is largely consistent regardless of the input data types (i.e., GE or AS). However, the sex effects on GE and AS showed a much weaker correlation ($R = 0.21$, *Figure 2D*), consistent with our earlier results (*Figure 2A, B*). We also found that age was a more vital driver of AS variation than GE in specific tissues like coronary artery and stomach, whereas sex contributed more to AS variations in whole blood, skeletal muscle, and most brain regions.

To further examine the sex-differential age effect, we compared the age-pcSVR between females and males (*Figure 2E, F*; GE and AS, respectively). Consistent with earlier findings (*Figure 2C*), we found significant age effects in both sexes across most tissues. Moreover, the age effects in both GE and AS were apparently larger in males than in females (*Figure 2E, F*). These results suggested that age and sex contributed significantly to transcriptome variation in various tissues with certain degrees of heterogeneity for different data types.

## Differential GE and AS analyses reveal general interactions of age-by-sex

We next identified specific genes or AS events responsible for sex or age effect on transcriptomic variations via a linear regression model containing an age-by-sex coefficient term (see Materials and methods, *Equations 4 and 5*). In each tissue, we identified the sex- or age-differential genes and AS events (*Figure 3—figure supplement 1A, B*). Consistent with previous reports (*Lopes-Ramos et al., 2020*; *Melé et al., 2015*; *Seidler et al., 2010*), the breast tissue contains the largest amount of sex-differentially expressed genes, while the age-biased genes were widely discovered in brain regions and artery-related tissues. Interestingly, in most tissues, there were no significant overlaps between age-differential genes and AS events (*Figure 3—figure supplement 1A*, hypergeometric test p-value <0.01). In contrast, we found significant overlaps between sex-differential genes and AS events in most tissues (*Figure 3—figure supplement 1B*).

As a control, we constructed a separate regression model to evaluate the contribution of sampling biases, including sample sizes and the total number of genes or AS events detected in each tissue (*Equation 6*). We found that the sample size did not significantly affect the number of identified sex- or age-differential genes/AS events (*Figure 3—figure supplement 1C, D*), suggesting our analyses are not affected by sampling biases. Furthermore, using the coefficient of the age-by-sex term in this

model allowed us to identify thousands of genes and AS events affected by the functional interactions between sex and age (i.e., the sex effects on GE/AS depend on the age, or vice versa) (*Figure 3—figure supplement 1E*), suggesting extensive sex-dimorphic changes of the transcriptome during aging.

A recent study reported the environmental effect on human genetic background during aging (*Balliu et al., 2019*), which should be considered when analyzing large datasets from different populations. In the linear regression model, we included the surrogate variables (SVs) as confounders to control for the effect of genetic backgrounds (*Equations 4 and 5*). Using the differential GE analysis in whole blood as an example (other tissues have the same trend), we evaluated the correlation between SVs with donors' ethnicity using Point Biserial Correlation and found that many SVs showed significant correlations with the donors' ethnical background (*Figure 3—figure supplement 2A*), suggesting that these SVs could effectively reflect the variables of genetic background. Additionally, we calculated the correlation between SVs with the top 5 principal components as judged by whole-genome sequencing (WGS) data (*Figure 3—figure supplement 2B*), and found roughly the same set of SVs also showed a significant correlation with PC1.

Moreover, recent studies demonstrated the powerful impacts of post-mortem interval (PMI) and time of death (TOD), which included the death seasons and day times on GE (*Ferreira et al., 2018*; *Wucher et al., 2023*). Thus, we carefully evaluated whether both factors are controlled as potential confounders in our linear model. Our results showed that PMI and TOD significantly correlate with covariates in most tissues (*Figure 3—figure supplement 2C–E*), suggesting that their effects could be effectively regressed in our model. Together, the SVs in the linear regression models could effectively control the confounders of human genetic regulation.

To further evaluate the robustness of the differential analyses, we performed detailed assessments of the linear regression models and the selection of GE/AS cutoffs using the decision-related brain region as an example. We calculated the fraction of original differential genes/AS events identified in each permutation iteration and the false discovery rate (FDR) for each gene/AS event using the permutation analysis in the linear regression model (see Materials and methods). Briefly, we generated the permuted data 1000 times by randomly shuffling the group labels (sex and age) with the same sample sizes as the original labels. The distribution of the p-values of sex/age-differential genes/AS events identified based on the original and each shuffled phenotype data was evaluated (*Figure 3—figure supplement 3A, D*). We first observed low fractions of sex/age-differential genes/AS events in most iterations under our cutoffs (*Figure 3—figure supplement 3*). We found that FDRs of most sex/age-differential genes/AS events were not more than 5% using the linear regression model (*Figure 3—figure supplement 2C, F*), which is much lower than the FDR of DESeq2 and edgeR (*Li et al., 2022*). As a result, this approach could justifiably control the false discovery for differential analysis. Furthermore, the TPMs of most sex- or age-differential genes were higher than 1, and the junction read counts (JC) of most differential AS events were higher than 10 (*Figure 3—figure supplement 4A, B*), suggesting that our cutoffs (TPM >1 or JC >10) preserved most of the differential GE and AS events. In addition, we found a low overlap between the genes with age-differential GE and AS under most combinations of TPM and JC cutoffs in the decision-related brain region (*Figure 3—figure supplement 4C*), which is consistent with our earlier findings (*Figure 3—figure supplement 1A*). Collectively, these controls confirmed the reliability of our differential analyses.

## AS regulation shows a stronger sex dimorphism during aging than GE across all tissues

We further studied the transcriptomic changes during the aging process in different sexes by identifying age-associated genes or AS events separately in females and males (*Supplementary file 3*). We found that the age-associated genes significantly overlapped between females and males in most tissues (p-values <0.01 by hypergeometric tests, *Figure 3A*), whereas the overlaps of the age-associated AS events were much smaller (*Figure 3B*). Consistently, in most tissues, the age effect on GE was more correlated between females and males (Spearman's correlation between $\beta_F$ and $\beta_M$, *Equation 7* than the age effect on AS *Figure 3C*), regardless of different GE/AS filters used in such analysis (*Figure 3—figure supplement 4D*). Taken together, these observations suggested that the age effect on GE is more consistent in both sexes, while age-associated AS showed a stronger

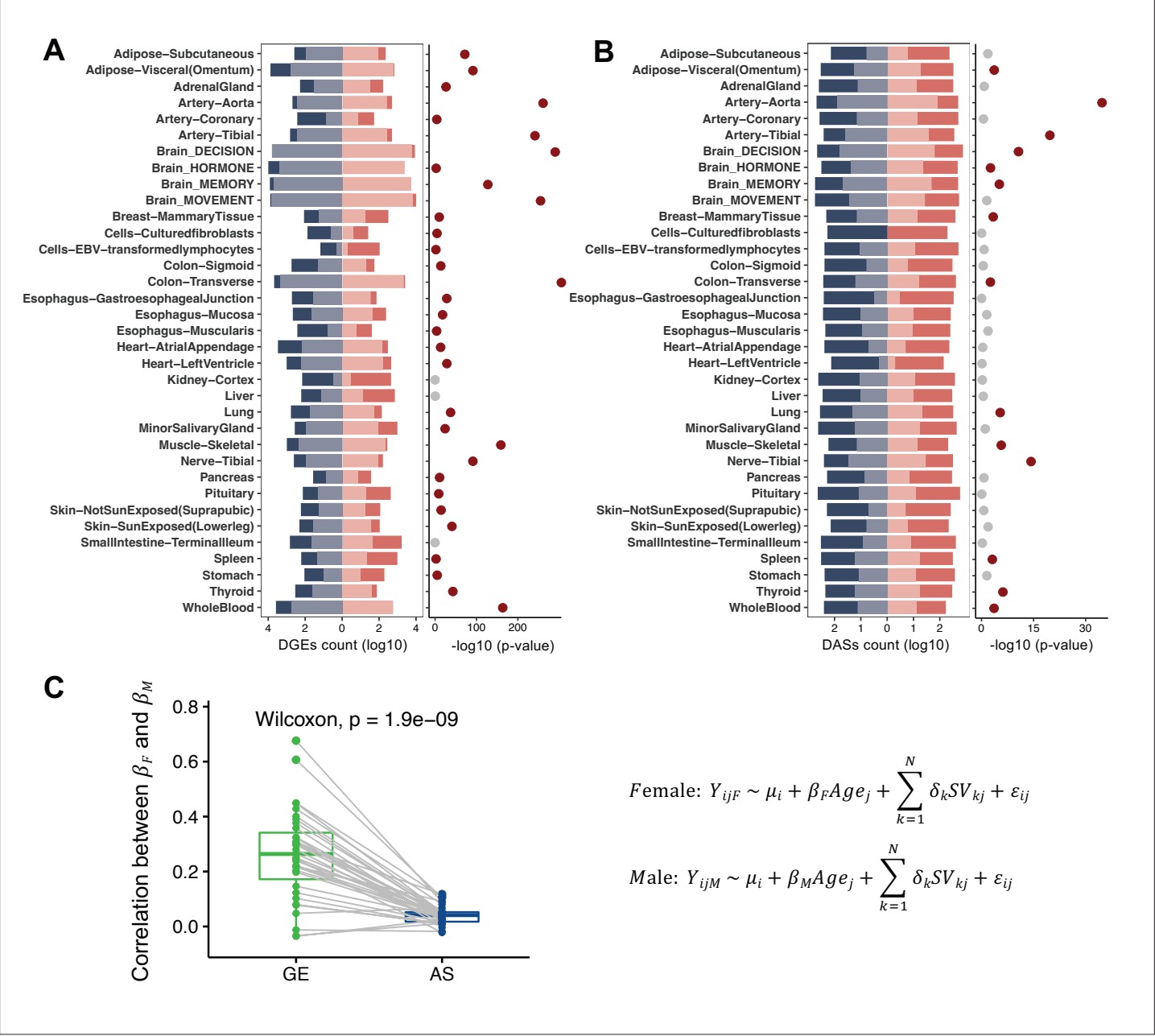

**Figure 3.** Sex-stratified age-associated genes and alternative splicing (AS) events across tissues. The numbers of age-associated genes (**A**, left) and AS events (**B**, left) in each sex. The red bars indicate the numbers of age-associated genes or AS events in females, while blue bars indicate those in males. The numbers of age-associated genes and AS events common in both sexes are shown in bars with relatively lighter colors. The significance of the overlapped age-associated genes (**A**, right) or AS events (**B**, right) between sexes is estimated using the hypergeometric test. Tissues with significant overlapped p-values are labeled by dark red dots. (**C**) Correlation between the age effects on genes (green) or AS events (blue) in females vs. males. The Y-axis represents the correlations of the effect sizes of age ($\beta_F$, $\beta_M$) between two sexes across all genes/AS events (Spearman's correlation). Gray lines link a pair of tissue in gene expression (GE) and AS, and the median ±1.5xIQR are plotted in box plot. The p-values are estimated using the Wilcoxon signed-rank test.

The online version of this article includes the following figure supplement(s) for figure 3:

**Figure supplement 1.** Differential gene expression (GE) and alternative splicing (AS) analysis to identify sex/age-differential genes and AS events.

**Figure supplement 2.** The evaluation of the confounding factors in the linear regression model.

**Figure supplement 3.** Permutation analysis of the linear regression model in differential analysis.

**Figure supplement 4.** Robustness of gene expression (GE) and alternative splicing (AS) cutoffs.

discrepancy between two sexes, implying that splicing regulation had a stronger sex dimorphism during aging.

## AS changes show a male-biased association with Alzheimer's disease in human brain

Because most age-associated AS events were sex-specific, we defined the AS events significantly affected by age in only one sex as the sex-biased age-associated AS events (sBASEs), and further examined their functional impacts on human health using Gene Ontology and Disease Ontology analyses. The functional association of sBASEs varied considerably in different tissues and sexes (*Figure 4—figure supplement 1*, *Supplementary file 4*). Having established patterns across all tissues, we next focused on tissue-specific AS changes in the brain that is highly affected by age-related processes. Since many neurological conditions show sex differences, the sex-biased splicing events in the brain could be helpful to explain differential susceptibility to age-related cognitive decline and neurodegeneration. Notably, the sBASEs in brain regions were significantly enriched in pathways such as amyloid-beta formation and cytoskeleton (*Figure 4A*), with the male decision-related brain region being particularly associated with cognitive and psychotic disorders (*Figure 4—figure supplement 2A*). Because previous studies suggested that the disease manifestations of Alzheimer's disease (AD) showed sex differences in cognitive decline and brain atrophy (*Ferretti et al., 2018*), we next focused on sBASEs in brain to explore their contribution to the sex-dimorphic risks of neurodegenerative diseases. This question was addressed by an integrative analysis of the large datasets from other independent AD studies, which is a common strategy (and strength) of computational biology in the big data era.

We analyzed the RNA-seq dataset of the brain prefrontal cortex with a dataset of 91 AD patients and healthy controls manually curated to match the sex and disease ratio (Female:Male ≈ 1:1; AD:Control ≈ 1:1, see Materials and methods). Surprisingly, although the number of sBASEs in males was smaller compared with that in females, we found that the AD-related AS events significantly overlapped with the sBASEs in males (p = 0.037) but not in females (p = 0.425) (*Figure 4B*). In addition, the magnitude of AD-associated AS changes across the male-specific sBASEs (i.e., $PSI_{AD} - PSI_{Ctrl}$) was significantly correlated with the age-associated AS changes in males (i.e., $PSI_{Old} - PSI_{Young}$) (Pearson's product-moment correlation, $R = 0.27$, $p = 6.6 \times 10^{-5}$), and such correlation was significantly higher than that in a separate analysis in females using the female-specific sBASEs ($R = 0.11$, $p = 0.04$; *Figure 4C*; see Materials and methods). In contrast, we found a significant overlap between the AD-related genes and the sex-biased age-associated GE changes in females (p = 0.031) but not in males (p = 0.60) (*Figure 4—figure supplement 2B*), supporting a recent finding that females are more vulnerable to AD during aging due to the post-menopause activation of C/EBPβ–AEP/δ-secretase pathway by sex hormone changes (*Xiong et al., 2022*).

In addition, we performed similar analyses using an independent dataset from ROSMAP containing 271 filtered RNA-seq samples from dorsolateral prefrontal cortex (DLPFC) with matched age distribution between AD and control samples. As expected, we found that the AD-associated AS changes of sBASEs in males showed a significantly stronger positive correlation with the age-associated changes ($R = 0.509$, $p < 2.2 \times 10^{-16}$) than such correlation across the sBASEs in females ($R = 0.371$, $p < 2.2 \times 10^{-16}$) (*Figure 4—figure supplement 2C*), which is largely consistent with the earlier finding using a smaller dataset (*Figure 4C*). This male-biased association indicated that the AS changes during aging could contribute more to the AD in males than females, suggesting additional molecular complexity in the sex effect of AD.

To further examine the contribution of individual sBASE to AD risk, we constructed a sex-stratified random forest classifier using a subset of sBASEs that were also detected in ROSMAP data for AD prediction. For each sex, we randomly divided the datasets of AD patients and controls into separate training and test sets and repeated the prediction 100 times (*Figure 4D*). For each selected sBASE, we also evaluated its averaged feature importance (i.e., mean decrease accuracy, MDA) across 100 iterations and the association with the classic AD neuropathology (such as tangles and plaques), resulting in the identification of a series of key sBASEs significantly associated with AD in a sex-specific manner. For example, the skipping of exon 10 in SLC43A2 negatively correlated with the neuritic plaques stage in males but not in females (*Figure 4E*). Meanwhile, skipping of exons 3 and 4 in FAM107A, a gene related to synaptic and cognitive functions, showed a male-specific negative

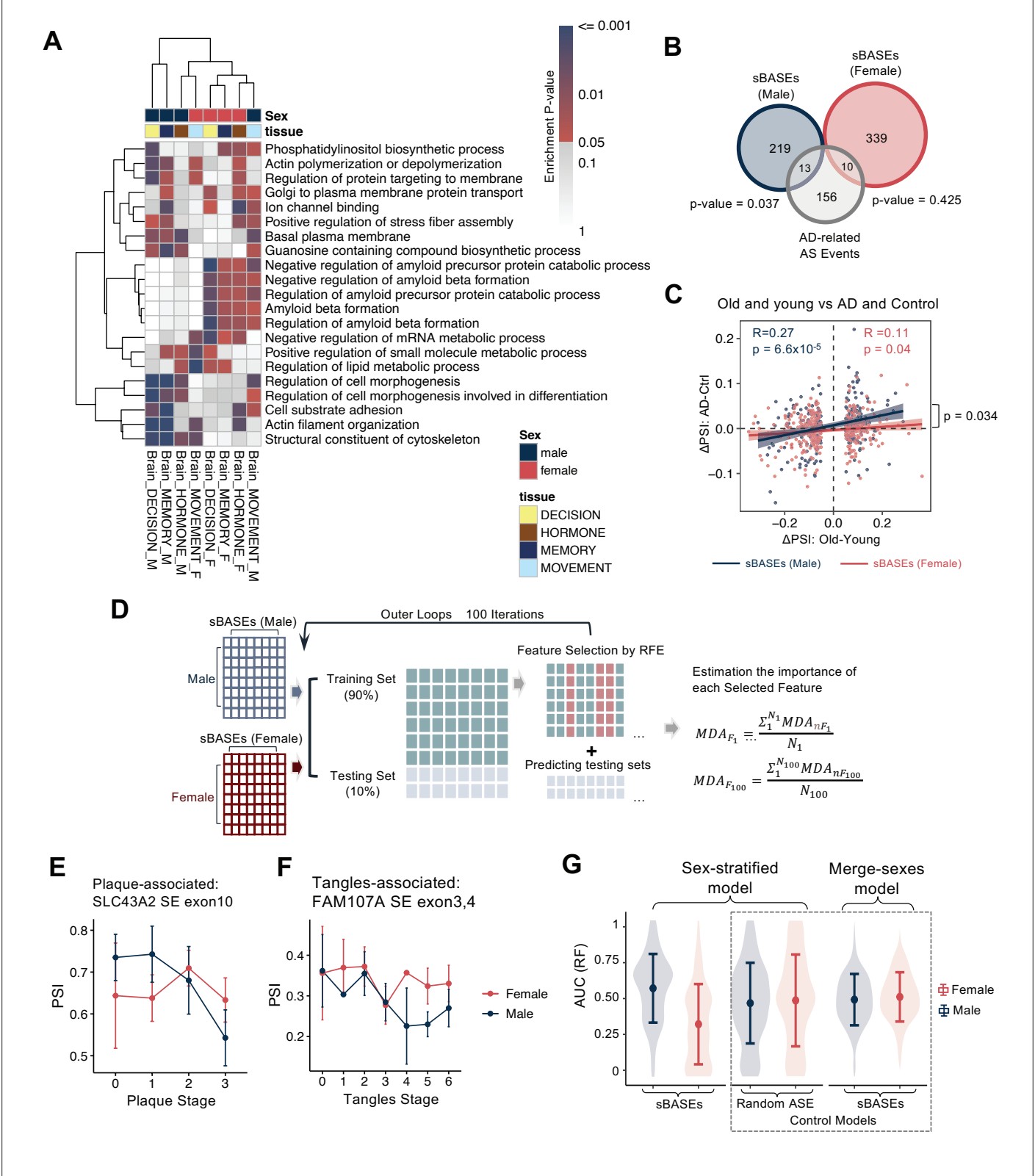

**Figure 4.** Male-biased associations between the alternative splicing (AS) changes during aging and Alzheimer's disease (AD). (**A**) GO analysis of the sex-biased age-associated AS events (sBASEs) in each sex and brain region. The heatmap shows the sex-specific pathways that are significantly enriched in more than three brain regions. Clustering is conducted by default parameters in *pheatmap* functions. The −log₁₀ transformed enrichment p-values are shown in the color scale. (**B**) Venn diagram between the sBASEs in females and males with AD-related AS events. The p-values are calculated

*Figure 4 continued on next page*

*Figure 4 continued*

using the hypergeometric test. (**C**) Correlation between AD- and age-associated AS changes in females and males. The *X*-axis indicates the $PSI_{Old} - PSI_{Young}$, while the *Y*-axis indicates $PSI_{AD} - PSI_{Control}$. sBASEs in females are labeled in red, while sBASEs in males are in deep blue. The estimated Rho and p-value by Spearman's correlation test in each sex are labeled on the top. (**D**) Model for AD prediction and feature importance evaluation. sBASEs are used for predicting AD in females and males, respectively. 90% of samples are randomly selected as training sets for 100 iterations. The recursive feature elimination approach is used for feature selection. Feature importance is evaluated by the averaged mean decrease accuracy (MDA) across 100 iterations. (**E**) AS levels of the skipped exon 10 on SLC43A2 during plaque stages in males and females. (**F**) AS levels of the skipped exons 3 and 4 on FAM107A during tangles stages in males and females. Mean ± standard error is shown in the error bar. (**G**) Performances of sex-stratified and merge-sexes models predicted by sBASEs or randomly selected AS events for 100 iterations. The control models (i.e., the sex-stratified model trained by randomly selected AS events and the merge-sexes model trained by sBASEs in each sex) are highlighted with dashed lines. The median ± 1.5xIQR of area under the curve (AUC) across 100 iterations is shown in the boxplot.

The online version of this article includes the following figure supplement(s) for figure 4:

**Figure supplement 1.** Functional enrichment of sex-biased age-associated alternative splicing events (sBASEs) across tissues.

**Figure supplement 2.** Sex-biased associations between sex-biased age-associated alternative splicing events (sBASEs) and diseases in decision-related brain region.

association with neurofibrillary tangle stages (*Figure 4F*). In addition, the prediction models based on sBASEs performed significantly better in males than females (*Figure 4G*, *Figure 4—figure supplement 2D*, left panel), suggesting that age-associated AS events could serve as a better predictor for AD risk in male patients. As negative controls, we performed the same analysis to predict male and female patients separately using randomly selected AS events (sex-stratified model; *Figure 4G*, middle panel), and predict all patients with male- or female-specific sBASEs (merge-sexes model; *Figure 4G*, right panel). We found that both control models showed no differences in males and females, indicating the crucial roles of sBASEs in AD during male aging. Collectively, our analyses of human brains revealed a surprising sex-biased association between RNA splicing and AD, suggesting that regulation of AS may play a critical role in the sex-dimorphisms of age-related diseases such as AD.

## sBASEs are regulated by splicing factors with sex-dimorphic expression during aging

AS is generally regulated by splicing regulatory *cis*-elements that specifically recruit various *trans*-acting splicing factors (SFs) to promote or inhibit splicing of adjacent exons (*Wang and Burge, 2008*; *Matera and Wang, 2014*). Dysfunction of the SFs has been reported as a hallmark of aging, presumably by regulating the genes in cellular senescence during the aging process (*Bhadra et al., 2020*). Consistently, our analyses identified a large number of age-associated SFs in human brain (*Supplementary file 5*). To further examine the sex-dimorphic regulation through age-associated SFs, we constructed an SF–RNA regulatory network between age-associated SFs and sBASEs in each sex. This process integrated the correlations between the SF levels and PSI of sBASEs, the significant changes of sBASEs upon SF knockdown, and the existence of SF-binding motifs around sBASEs (*Figure 5A*, left panel). We required the regulatory connections between the SFs and AS events that pass all these three criteria. Such regulatory networks can be generated in different tissues, and we focused on the brain due to its association with age-related diseases such as cognitive disorders and synaptic diseases.

We identified several key age-associated SFs that regulate multiple sBASEs in decision-related brain regions (*Figure 5A*, right panel). Interestingly, several SFs showed significant age association only in females (e.g., TIA1 in *Figure 5B*, left panel), while others were associated with age in both sexes (e.g., FXR2 in *Figure 5B*, right panel). This finding is consistent with our earlier observation that more sex-specific age-associated AS events were identified in decision-related brain regions of females than males (*Figure 4B*). Most of the sBASEs were regulated by sex-specific age-associated SFs rather than the SFs associated with aging in both sexes (*Figure 5C*). Functional enrichment analysis showed that these sBASEs regulated by female-specific age-associated SFs were significantly enriched in glial cells and oligodendrocyte development (*Figure 5D*), which participated in the myelin generation of aging brain and neurodegenerative diseases in central nervous system (*Bergles and Richardson, 2016*). These results indicated that the SFs with sex-dimorphic expression during aging,

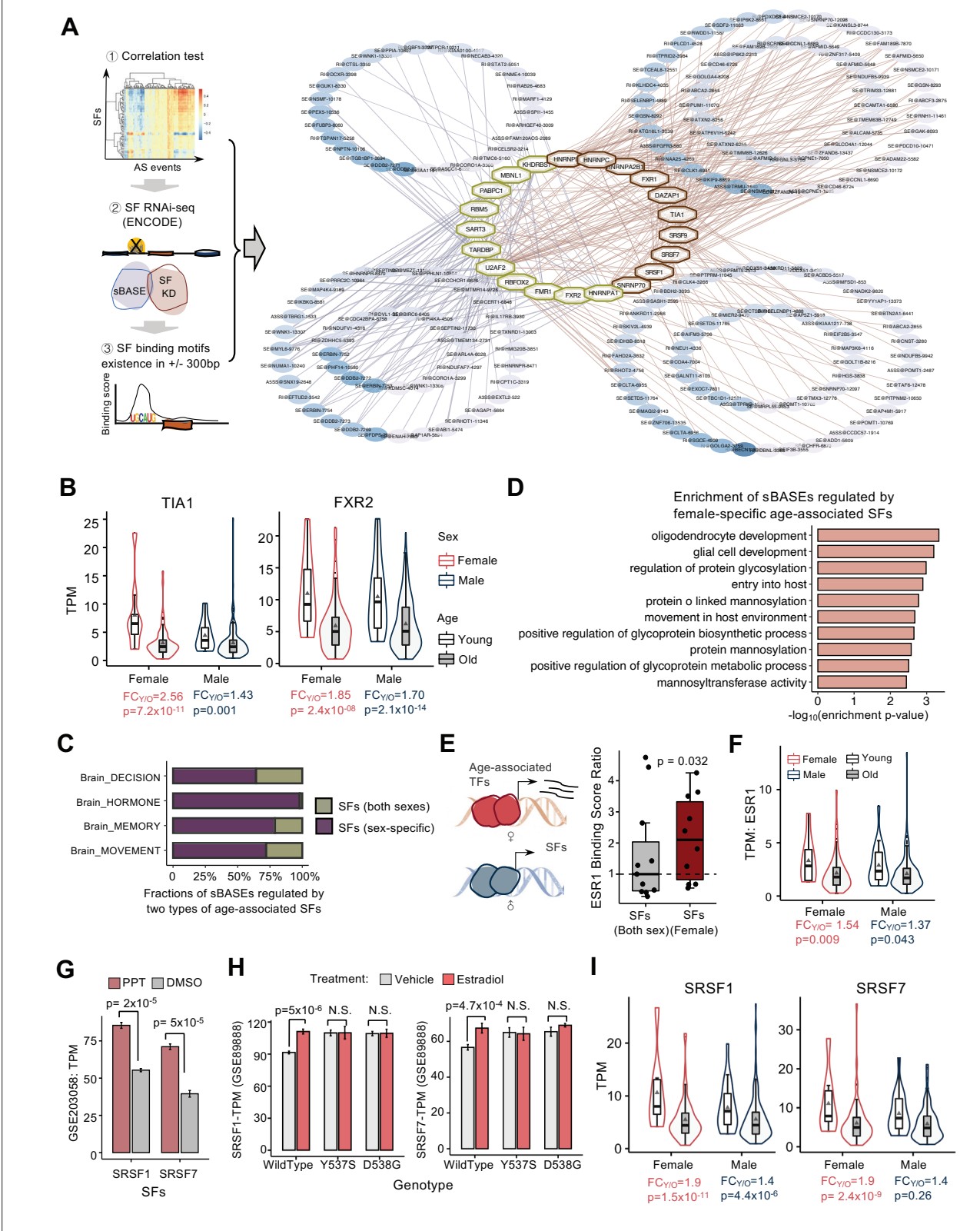

**Figure 5.** Sex-dimorphic alternative splicing (AS) regulation during aging in decision-related brain region. (**A**) Schematic diagram for constructing AS regulatory networks (left) and the networks of the decision-related brain region (right). The blue and red lines indicate the regulations in males and females. Hub genes in octagons are splicing factors (SFs). The red ones are female-specific age-associated SFs, while the age-associated SFs common in both sexes are labeled in yellow. The color of the ellipse indicates the significance of sex-biased age-associated AS events (sBASEs) during aging, and

*Figure 5 continued on next page*

*Figure 5 continued*

the thickness of the line shows $-\log_{10}$ Spearman's correlation p-values between the TPMs of SFs and PSIs of AS events during aging. (**B**) Examples of female-specific age-associated SFs (left) and age-associated SFs common in both sexes (right) during aging. Median ±1.5xIQR is shown in the boxplot. The grey triangles indicate averaged expression level. The p-value and fold change between old and young are labeled at the bottom. (**C**) Percentages of the sBASEs regulated by the sex-specific age-associated SFs in four brain regions. (**D**) Functional enrichment of sBASEs regulated by female-specific age-associated SFs based on MsigDB. (**E**) Schematic diagram for the transcriptional regulation on age-associated SFs via nuclear receptors (left). The boxplot shows the ESR1-binding scores on age-associated SFs (right). The *Y*-axis indicates the ratio of ESR1-binding scores of each SF divided by the median binding score of age-associated SFs common in both sexes. (**F**) Expression levels of ESR1 during aging. The legends of each group are the same as those in B. (**G**) Expression levels of SRSF1 and SRSF7 treated by estrogen receptor agonist (PPT) vs. DMSO in MCF-7 cell line (*N* = 3). (**H**) Expression levels of SRSF1 and SRSF7 treated by 1 nM estradiol (E2) vs. vehicle control (veh) in ESR1 wild-type, Y537S, and D538G mutant MCF-7 cells (*N* = 4). The error bars indicate the mean ± s.d. (**I**) Expression levels of SRSF1 and SRSF7 during aging. *Y*-axis indicates the normalized TPM values. The p-value and fold change between old and young groups in each sex are labeled at the bottom. The legends of each group are the same as those in **B**.

The online version of this article includes the following figure supplement(s) for figure 5:

**Figure supplement 1.** Gene expression regulation of sex-biased age-associated splicing factors by estrogen via ESR1 in multiple datasets.

especially the female-specific changes of SFs, were one of the main factors in regulating sBASEs with important biological functions.

Furthermore, we studied the potential mechanism that controls the female-specific changes of SFs during aging in decision-related brain regions. It was previously reported that the female sex hormone estrogen consistently decreases during lifespan (*Ober et al., 2008*; *Horstman et al., 2012*) thus, we speculated that estrogen may play roles in controlling transcription of SFs during aging through activation of nuclear receptors. To this end, we analyzed the public ChIP-Atlas data to examine the binding of canonical estrogen receptor ESR1 to the promoter region. We found that the ESR1-binding ratios on the female-specific age-associated SFs were significantly higher than the median value of the age-associated SFs common in both sexes (*Figure 5E*; p = 0.032), suggesting that ESR1 was more likely to bind the promoters of the female-specific age-associated SFs. In addition, the age-associated decline of ESR1 was more substantial in females than males (*Figure 5F*), and some of the female-specific age-associated SFs (e.g., SRSF1 and SRSF7) could be stimulated by propyl pyrazole triol, an agonist of ERα (*Figure 5G*; *Cao et al., 2024*).

These observations were further confirmed by an independent experiment where the wild-type and two constitutively active mutants (Y537S and D538G) of ESR1 were engineered into MCF-7 cells using CRISPR–Cas technology (*Bahreini et al., 2017*). The resulting cells were treated with estradiol or vehicle controls and subjected to transcriptome profiling by RNA-seq, and the data were reanalyzed using the same pipeline. We found that the basal levels of SRSF1 and SRSF7 were significantly higher in the cells with constitutively active ESR1 (*Figure 5H*, comparing samples treated with vehicle controls), and SRSF1 and SRSF7 were further induced by the estradiol treatment only in cells with wild-type ESR1 but not in cells with mutated ESR1 (*Figure 5H*, compare the samples treated with estradiol vs. control in different cells), indicating that the activation of the ESR1 pathway can promote expression of SRSF1 and SRSF7. The similar changes were also found in other SFs including HNRNPA2B1 and HNRNPC (*Figure 5—figure supplement 1A*). As expected, the expression levels of SRSF1 and SRSF7 showed an obvious decrease during female aging (*Figure 5I*).

To further confirm our findings in different tissues, we analyzed the RNA-seq data with ER knockout and estrogen treatments in mouse brain arcuate nucleus (ARCs) (*Yang et al., 2017*) and found a consistent trend in the expression changes for SFs Srsf1, Srsf7, and Hnrnpa2b1 under estradiol (E2) regulation (*Figure 5—figure supplement 1B*). Specifically, these SFs were activated by estrogen in wild-type mice but exhibited reduced expression in ER knock-out mice. The direct binding of estrogen was also examined through integrating the ESR1 ChIP-seq data (*Hewitt et al., 2014*). As expected, a major ESR1-binding peak was found in the promoter regions of Srsf7, which was increased by estrogen treatment (*Figure 5—figure supplement 1C*).

Collectively, these analyses across multiple datatypes suggested that the female-specific SFs (such as SRSF1 and SRSF7) are more sensitive to the estrogen changes via ESR1-mediated pathway, leading to extended sex-biased AS regulations during aging.

## The aging rates of GE show sex dimorphism during aging across multiple tissues

In the earlier results (*Figure 3A*), we found a large proportion of age-associated genes are common in both sexes. Therefore, we asked whether the levels of GE change at the same pace during aging in females and males. The age-associated genes ubiquitously changed in both sexes were analyzed for consistency. We first used the Autoregressive Integrated Moving Average (ARIMA) model (*Brockwell et al., 2016*) to capture the age-associated genes with significant chronological changes beyond random fluctuations, which are the main contributors to transcriptome dynamics during aging. We further quantified the GE changes between adjacent age windows using breakpoint analysis (*Márquez et al., 2020*; *Figure 6* see Materials and methods). Such analyses were conducted across all tissues in both sexes (*Figure 6—figure supplement 1A*), with two tissues shown in (*Figure 6*). A higher value on the *Y*-axis (mean −log *P*) indicated a larger GE change at the given age, and thus the peaks were regarded as aging 'breakpoints' that represent the key time point of transcriptome aging. Compared to females, males showed significantly larger GE changes during aging in most tissues (*Figure 6— figure supplement 1A*), except transverse colon, fibroblast cells, and whole blood (*Figure 6—figure supplement 1B*). The number of breakpoints also showed sex-dimorphism in several tissues. For example, in sun-exposed skin and colon, males showed two main breakpoints around ages 35 and 50, whereas females only have one breakpoint at age ~45 (*Figure 6—figure supplement 1A*).

Moreover, we found that the major aging breakpoints (i.e., breakpoints with the largest GE changes) occurred earlier in males than females across most tissues, and the magnitudes of changes were also more prominent in males (*Figure 6C*), suggesting that males may age faster with a larger age-associated GE change. In addition, we conducted breakpoint analysis with all chronological genes in both sexes and found similar results when comparing the breakpoints in males and females (*Figure 6—figure supplement 1C*). We also performed a similar analysis using chronological sBASEs and all chronological AS events in females and males separately, and again found an earlier and more obvious breakpoint (*Figure 6—figure supplement 1D, E*). In summary, these results uncovered an earlier aging and a higher aging rate in males at molecular levels, again implying that males could be surprisingly more vulnerable to aging.

## Aging contributors in separate sexes show distinct functions

To further examine how the individual gene affected aging rates, we designed a disturbance analysis to identify aging-modulated genes (AMGs) (*Figure 6*). Specifically, we randomly removed 20% of chronological genes and repeated the breakpoint analysis with the remaining genes to test if the breakpoints were significantly altered (see Materials and methods). This process was repeated 200 times, and the genes enriched in the 200 random samplings (judged by Fisher's exact test p < 0.05) were defined as AMGs that serve as the key contributors to molecular aging.

We identified a total of 1035 AMGs across different tissues (*Supplementary file 6*), around 70% of which (723 in 1035) were sex-specific (*Figure 6*). Functional enrichment analysis suggested that the female-specific AMGs were significantly enriched in cellular respiration and NADH dehydrogenase pathways (*Figure 6F*), which is consistent with an earlier report of their roles in human aging and longevity (*Durieux et al., 2011*). Such functional enrichment was not found in the male-specific AMGs (*Figure 6G*), implying that female aging may be more sensitive to mitochondrial and oxidative functions. Intriguingly, we also found many male-specific AMGs relevant to myeloid leukocyte differentiation, phagocytosis, and macrophage apoptotic processes, implying a sex-differential role of the immune system during aging.

To determine the potential mechanism of the sex-dimorphic aging rate, we focused on sex hormones that are critical for transcriptional regulation. We found that the androgen receptor (AR) tends to bind the promoters of male-specific AMGs (p = 0.007), suggesting potential regulations by androgen during male aging (*Figure 6G*, top panel). On the other hand, female-specific AMGs were more likely to be bound and regulated by estrogen receptor ESR1 (*Figure 6H*, bottom panel), suggesting the potential regulation by distinct sex hormones.

Furthermore, we examined the detailed expression profile of AMGs from the whole blood that is commonly used for health surveillance. The AMGs common to both sexes can be grouped into three clusters based on age-associated expression profiles. GE of the largest cluster increased at young ages and abruptly decreased in mid-ages in males, but monotonically decreased in females

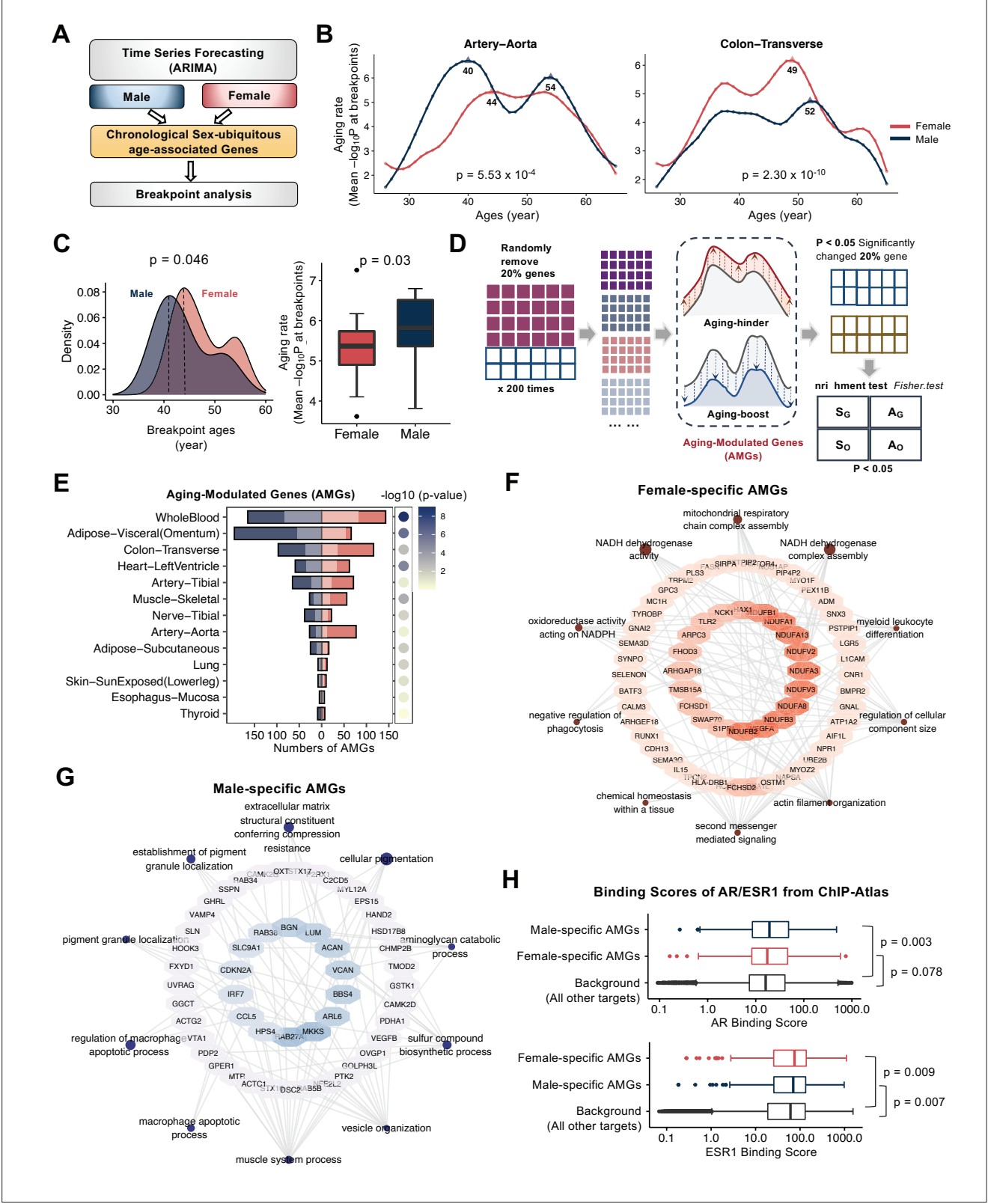

**Figure 6.** Sex-dimorphic aging rate of gene expression (GE) during aging process. (**A**) Workflow of time series and breakpoint analysis of GE in each sex. (**B**) Aging rates and breakpoints in the aorta (left) and transverse colon (right). The Y-axis shows the estimated aging rate (see Materials and methods). The p-values are calculated using the Wilcoxon signed-rank test. (**C**) Characteristics of the major breakpoints across tissues in each sex. The left panel shows the distribution of the major breakpoints across multiple tissues. The right panel shows the aging rate at the major breakpoints. The

*Figure 6 continued on next page*

*Figure 6 continued*

p-values are calculated using the Wilcoxon signed-rank test. (**D**) Pipelines for identifying aging-modulated genes (AMGs) in each sex and tissue (see Method). (**E**) AMGs in different sexes and tissues. The numbers of AMGs in females (red) and males (blue) are shown in the bar plot (left). Tissues are ordered by the number of AMGs common in both sexes, which are shown in lighter colors. Significances of the overlapped AMGs are calculated using hypergeometric test and colored in a gradient palette (right). GO analysis of male-biased AMGs (**F**) and female-biased AMGs (**G**). Sex-specific GO terms with top 10 $-\log_{10}$ transformed enrichment p-values are plotted. GO terms are labeled by purple dots, the size of which indicates the significance of the enrichment analysis. The larger the dots, the more significant enrichment scores of the biological pathways. AMGs are labeled by blue dots, the color of which indicates the number of GO terms associated with those genes. (**H**) The binding scores of AR (top) and ESR1 (bottom) on male-specific AMGs, female-specific AMGs, and randomly selected AR/ESR1 target genes. The p-values are calculated by the Wilcoxon rank-sum test. The error bars indicate median ±1.5xIQR.

The online version of this article includes the following figure supplement(s) for figure 6:

**Figure supplement 1.** Breakpoint analysis across multiple tissues.

**Figure supplement 2.** Gene expression (GE) patterns of the aging-modulated genes (AMGs) in whole blood tissue.

(*Figure 6—figure supplement 2A, B*). These genes were functionally enriched in neutrophil and granulocyte migration and RNA splicing (*Figure 6—figure supplement 2C*). Moreover, we also examined sex-specific AMGs and found very distinct expression patterns (*Figure 6—figure supplement 2D, E*). Interestingly, while most of these genes showed decreased expression at old ages, the expression of some male-specific AMGs gradually increased during aging (*Figure 6—figure supplement 2D*, dotted box). Taken together, these results suggested that these age-associated genes modulating the transcriptome aging rate showed different functional impacts in females and males.

## Discussion

Previous studies in selected tissues suggested that sex is a key factor in affecting human aging and age-related diseases (*Sampathkumar et al., 2020*; *Hägg and Jylhävä, 2021*), which may contribute to different life expectancies between two sexes. However, few pan-tissue investigations on age-by-sex effect were conducted at the transcriptome level, especially in the resolution of AS isoforms. By comprehensively analyzing ~17,000 samples across 54 tissues, we uncovered individual sex or age effect on global transcriptome variations, as well as the combined age-by-sex effects. Using a sex-stratified differential analysis, we found that AS changes showed a stronger discrepancy between females and males during aging. We further identified the regulatory SFs for sBASEs and explored the underlying mechanism of sex-differential AS regulation during aging. Conversely, we found the change of age-associated GE was consistent between two sexes, with a faster aging rate in males. Our analysis provided a comprehensive landscape of how sex and age affect human transcriptome, revealing the sex-dimorphic aging at a multi-tissue level.

Multiple mRNA isoforms can be produced from a single gene through AS, increasing the transcriptomic and proteomic complexity in different cells (*Djebali et al., 2012*; *Wang et al., 2008*). Therefore, analyses of AS changes during aging of each sex may provide additional information beyond the GE profile. Interestingly, we observed different patterns when analyzing the profiles of GE vs. AS. For example, judged by the AS changes, we observed more tissues with a significant sex or age effect on transcriptome and a higher discrepancy between females and males from the combinatorial effects of age-by-sex (*Figure 3*), highlighting the contribution of splicing in sex dimorphism during human aging at an isoform-specific resolution.

The sex-dimorphic patterns of AS and GE not only illuminate aging-related transcriptomic changes but also reveal intriguing connections to age-related diseases, such as AD. Most studies showed that females have a higher risk of AD due to the post-menopause changes of sex hormones (*Xiong et al., 2022*) however, a higher incidence rate of mild cognitive impairment in males is also reported (*Roberts et al., 2012*). Additionally, based on an early study on a large cohort of 17,127 participants from UK Biobank, age-related cognitive risk factors are more widely discovered in males, suggesting a potential vulnerability of males to cognitive decline with age (*Foo et al., 2021*). We further validated our findings using the Alzheimer's disease dataset, ROSMAP, where the consistent correlations between aging-related splicing changes and AD-related changes were observed, providing additional evidence for the robustness of our results. At the splicing level, we surprisingly observed a male-biased association between age-associated AS changes and AD (*Figure 4*). Such associations could not be

detected by analyzing GE, suggesting that analysis at isoform resolution can provide different information into sex-dimorphic aging. We also designed a sex-stratified random forest classifier to predict AD in males and identify several key AS events contributing to such prediction. Due to the limited sample size, large variations were observed between different training/testing iterations during the sex-stratified AD prediction. However, we still observed a higher performance of the classifier in males compared to the control model, which is in concordance with that the AS changes during aging may contribute more to AD in males than females. Our findings suggest that male-biased age-associated AS changes may contribute to the increased vulnerability of males to cognitive decline, providing a complementary perspective to existing research. We expect that future investigations using bigger datasets will further identify the sex differences in AD.

Using a pan-tissues analysis, we discovered that transcriptomic aging happened earlier in males than in females (*Figure 6D*), consistent with the reported shorter life expectancy in males (*Hägg and Jylhävä, 2021*). Furthermore, we identified many sex-specific AMGs with different biological functions, such as the macrophage and mitochondria functions (*Figure 6F, G*). These genes tend to be regulated by the dominant sex hormones in each sex. The investigation of the aging rate provided insights into the intervention of the aging process or age-related diseases and, to some extent, made available for identifying therapeutic targets at specific age points. It is worth noting that our analysis of the aging rate in this section does not include the data from brain tissues due to the sample sparsity in selecting parameters.

Our analyses also have some technical limitations, the biggest of which may be caused by the cell type and sample heterogeneity between different donors of GTEx datasets. Since the GTEx dataset is the most comprehensive dataset with both transcriptome and WGS data, it is a good public resource for sex, age, and age-by-sex interaction analysis within a healthy context. Compared with the earlier studies that used either microarray or other RNA-seq data to analyze the effect of age or sex on brain transcriptome, our main findings supported the main conclusion of these reports, including the diverse gene profiles across brain regions during aging, as well as more vulnerability in males on a global scale (*Berchtold et al., 2008*; *Zhou et al., 2023*). In addition, GTEx has a very strict selection of healthy donors to reduce biases from different diseases (*Carithers et al., 2015*), which is a unique feature compared to other cohorts that often contain patients of various diseases that may affect the analyses at the molecular level. In particular, the large-scale brain samples provide a unique opportunity to analyze transcriptomic changes in sex-dimorphic aging. However, several technical challenges could limit the generalizability due to dataset-specific biases, including cell type heterogeneity, postmortem artifacts, as well as sequencing biases. For example, GTEx data is bulk RNA-seq, which does not capture cell-type-specific transcriptomic changes. Given the cellular complexity of the brain tissues, the observed differences in GE and splicing may be influenced by shifts in cellular composition rather than intrinsic transcriptional regulation.

The second limitation is the existence of potential confounding factors (like batch effect, genetic background, etc.) that are often associated with big data analyses. However, when such batch effects are entangled with the effect from distinct sex/age groups, it becomes technically challenging to effectively remove such factors without canceling out the contribution of sex/age, which is a common technical difficulty reported also in other analyses (*Nygaard et al., 2016*). We take several measurements to reduce the effect of confounding factors. First, we utilized the original TPM/PSI data to aggregate sex/age effects and conducted a permutation analysis to assign an empirical p-value to each pcSVR. This approach provided a robust statistical framework for evaluating the significance of sex/age effects while accounting for potential batch effects. Secondly, in the context of the differential analysis, we preserved all possible confounders and identified changes that were more obvious than the effects of these confounders, a strategy that reduces false positive discoveries originating from potential batch effects. Another technical limitation is related to the small number of samples in some tissues, and thus we either omitted them from aging analyses (like pituitary, bladder, and kidney) or combined similar tissues (like brain regions) for detailed analyses.

Moreover, we derived the AS regulatory networks using the multi-dimensional large-scale data from highly proliferative and easily cultivable cancer cell lines, which may not be an ideal model for the post-mitotic cells in human tissues. However, we dissected the RBP–RNA regulatory network by integrating multi-dimensional data obtained through several orthogonal state-of-the-art approaches (*Figure 5A*), including the eCLIP-seq, in vitro RNA affinity evaluation, RNA-seq with RBP depletion,

and chromatin association (*Van Nostrand et al., 2020*). In addition, the RBP–RNA-binding relationships seem to be universal, and the distinct splicing outcomes are mainly determined by the tissue-specific expression and activity of different splicing regulatory factors. Therefore, it is possible to use the data in cancer cells to derive a connectivity map of RBP–RNA, and then use the tissue-specific expression of RBPs to refine the regulatory map.

It was reported that there is only a moderate correlation between mRNA and protein abundances for many human genes, mainly because of the variation in translation efficiency of different mRNAs and the additional regulation in protein degradation. Here we constructed the SF–RNA regulation network with the shRNA-seq data that represents the reduction of specific SFs in both RNA and protein levels. Such data is more reliable than the correlation of cell-specific expression of RNA and protein, and thus it is feasible to evaluate the consequences of gene silencing by looking at the RNA level. To minimize tissue biases in the RBP–RNA regulation network, we incorporated three-step processes (see Materials and methods) including the restrictions of the correlations between RBP expression and AS events within each tissue. Additional large-scale experiments are needed in the future to globally profile the physical and functional interactions between RNA and RBP with a higher accuracy from different tissues.

Mechanistically, we found a prominent role of sex hormones in regulating sex-dimorphic aging or age-related diseases. Using a range of RNA-seq and ChIP-seq data, we uncovered a type of female-specific age-associated AS regulation (*Figure 5A*), which may be achieved partially through the estrogen-mediated regulation of female-specific SFs (*Figure 5E, I*). We further quantified chronological changes of transcriptome dynamics using breakpoint analysis and observed the sex-dimorphic aging rates across multiple tissues. Our analysis observed the non-linear aging patterns with two breakpoints, which is consistent with recent findings, with some differences in specific age points due to sex differences as well as tissue diversities (*Shen et al., 2024*). In addition, our aging breakpoints are largely consistent with the changes in circulating sex hormones during aging. The level of testosterone starts to decrease at the age of 35–40 in males, while in females, estrogen level starts to decrease at a later age (i.e., ~50 years old) (*Ober et al., 2008*; *Horstman et al., 2012*). These breakpoints could represent key junctures in the aging process that align with the non-linear patterns of aging and disease progression. Furthermore, we identified a series of female- and male-specific AMGs, whose functions are consistent with the reported roles of sex hormones. For example, estrogen was reported to participate in the antioxidant system via NADPH (*Ober et al., 2008*; *Horstman et al., 2012*), which is functionally enriched in female-specific AMGs (*Figure 6*). Conversely, androgen was reported to affect the inflammation in immune response (*Ober et al., 2008*; *Horstman et al., 2012*), which is enriched in the male-specific AMGs (*Figure 6G*). Consistently, we found that the sex-specific AMGs were more likely to be regulated by androgen receptor in males or estrogen receptor in females (*Figure 6H*). However, the regulation of sex-biased aging is probably more complicated. Other factors, including mitochondrial functions, genes encoded by sex chromosomes, non-coding RNAs, and transposable elements, may also contribute to the sex-dimorphism during aging (*Hägg and Jylhävä, 2021*; *Davis et al., 2021*; *Ramanujan et al., 2021*; *Yoshida and Apte, 2022*). In addition, it is possible that the genetic variants, including expression QTL (eQTLs) and splicing QTLs (sQTL), can affect GE or AS in a sex-biased or age-associated fashion (*Oliva et al., 2020*; *Balliu et al., 2019*; *Xiong et al., 2021*). Future studies of these factors will improve our understanding of the intricate regulation of transcriptome dynamics, providing us a broader horizon at a multi-omics scale.

Various biological aging clocks had previously been constructed based on different molecular markers, including DNA methylome (*Jylhävä et al., 2017*; *Horvath and Raj, 2018*; *Galkin et al., 2021*), transcriptome (*Meyer and Schumacher, 2021*), circulating immune proteins (*Sayed et al., 2021*), human gut microbiome (*Galkin et al., 2020*), and multi-omics features (*Li et al., 2023*). Such aging clocks generally pay little attention to the difference between females and males. However, our results advocate for a sex-specific age model. In addition, certain age-related diseases may accelerate the aging process, as tumors aged 40% faster than matched normal tissues based on DNA methylome (*Hannum et al., 2013*). Therefore, we speculate that constructing separate aging clocks in distinct sexes will help model the progression of age-related diseases accurately.

## Materials and methods

### Transcriptome quantifications and data processing

We used the transcriptome data from the Genotype-Tissues Expression (GTEx) project version 8.0, which contains 17,382 samples in 54 tissues (including 13 brain regions and two cell lines). We downloaded the RNA-seq bam files and the phenotype information from dbGaP (study accession: phs000424. v8.p2; table accession: pht002743.v8.p2.c1). The tissues with a small number of samples and specific to only a single sex (such as ovary and testis) were removed for a better comparison. In addition, for each donor, we combined 13 brain regions into 4 main functional regions, including hormone- or emotion-related region (amygdala, anterior cingulate cortex, and hypothalamus), movement-related region (cerebellar hemisphere, cerebellum, and spinal cord), memory-related region (caudate, hippocampus, nucleus accumbens, putamen, and substantia nigra) and decision-related region (cortex and frontal cortex). As a result, for the N donors that each have 13 brain regions ($N \times 13$ sample matrix), we merged the data into 4 functional regions to generate an $N \times 4$ sample matrix in the downstream analyses. In total, 16,202 samples in 35 tissues or brain regions (31 tissues and 13 brain regions) from 948 donors passed these filters for further analysis. The numbers of samples in each tissue are shown in *Supplementary file 1*.

GE and AS quantifications were performed by Paean (*Li et al., 2018a*), a parallel computing system on the GPU–CPU platform with high computational efficiency for super-large datasets. We calculated TPM (transcripts per million) for protein-coding genes and PSI (percent of spliced-in) values for AS events, including skipped exon (SE), alternative 5′ splice site (A5SS), alternative 3′ splice site (A3SS), and retained intron (RI). Genes with low expression (average TPM <1) were removed. We also filtered the AS events by first removing the samples with missing PSI values in more than 50% of AS events (16,182 samples left), and then filtered the AS events using the following criteria: (1) percentage of the missing PSI values in <5% samples; (2) averaged total reads counts in spliced junctions (i.e., spliced in counts + spliced out counts) >10; (3) non-constant PSI values across samples; (4) max(PSI) – min(PSI) >0.05; (5) standard deviation >0.01; (6) averaged PSI value in the range from 0.05 to 0.95; (7) TPM of the spliced genes >1. We focused on the AS events in protein-coding genes. In different tissues, the average numbers of genes and AS events for downstream analysis were 12,144 and 13,939, respectively.

Even after removing the AS events with too many missing values, the remaining AS data were still too sparse in some samples for a reliable statistical analysis. On average, there are about 0.17% AS events missing from the RNA-seq samples in different tissues. For a single RNA-seq sample in certain tissues, there were up to 15% AS events still missing (i.e., below the detecting limit). Several imputation techniques (reference-based and reference-free) have been widely used and easily implemented in multi-omics datasets (*Song et al., 2020*), and we used the *k*-nearest neighbors algorithm for missing PSI value imputation in this analysis.

### Principal component-based signal-to-variation ratio

To evaluate the sex/age effect on human transcriptome, we designed a pcSVR based on PCA and SNR. The PCA process could maximize the global transcriptomic variations, and SNR could calculate the distances between sex/age groups divided by the noises within each group (*Equations 1–3*, female vs. male and young vs. old). We selected the principal components that captured more than 80% of the global variation (other PC cutoffs capturing 50–90% of variations were also tested as controls). The SNR of the principal component score was calculated according to the following formula:

$$pcSVR\left(X, Y\right) = \frac{\|\bar{X} - \bar{Y}\|_2}{\sqrt{\dfrac{\sigma_X^2}{N_x} + \dfrac{\sigma_Y^2}{N_y}}} \tag{1}$$

$$\sigma_X^2 = \frac{\sum_{i=1}^{N_x} \|x_i - \bar{X}\|_2^2}{N_x - 1} \tag{2}$$

$$\sigma_Y^2 = \frac{\sum_{i=1}^{N_y} \|y_i - \bar{Y}\|_2^2}{N_y - 1} \tag{3}$$

where $X$ and $Y$ label two groups with sample sizes $N_x$ and $N_y$ (females and males; young and old); $x_i$ and $y_i$ are the PC scores in sample $i$; and $\bar{X}$ and $\bar{Y}$ are average PC scores across samples in each group calculated by GE or AS.

Here, to figure out whether the pcSVR is statistically significant enough, we conducted the permutation analysis and derived empirical p-values of the pcSVR. We first randomly selected subsamples (50% numbers of the minimum sample sizes of two groups) 10,000 times in each sex/age group and computed the averaged pcSVR. Then, we also randomly selected the same number of subsamples 10,000 times in each tissue regardless of the sex/age labels. The empirical p-values were calculated by the percentage of tested 10,000 pcSVR values (i.e., calculated using 50% sub-sampling irrespective of the group labels for 10,000 repeats) higher than the average pcSVR (i.e., calculated using 50% sub-sampling with the group labels for 10,000 repeats). The null hypothesis is that the two sets of pcSVR values are drawn from the same distribution. We performed all of the random sampling approaches using even probabilities in the R function 'sample' with the parameter prob = NULL. The tissues with more than three samples in each age group in females or males are included for this analysis (total 29 tissues and 4 brain regions).

To compare different approaches for empirical p-value estimation, a bootstrap with the replacement approach was also performed for null distributions. The process was repeated 10,000 times to minimize the biases of random sampling to construct a proper null distribution. Compared with the bootstrapping approach with replacement, the sub-sampling approach could also evaluate the robustness of pcSVR values to the sample size. Genes and AS events encoded by the protein-coding genes with averaged TPM >1 were considered. To evaluate the contribution of sex chromosomes to pcSVR, we also removed the genes and AS events encoded by sex chromosomes and recalculated the pcSVR with corresponding p-values in each tissue.

## Identification of differentially expressed genes and AS events

A recent study reported that DESeq2 and edgeR have unexpectedly high FDR in analyzing large-scale datasets, as well as the inconsistency between differentially expressed genes identified by these two approaches (*Li et al., 2022*). Linear regression models have been widely used for transcriptomic and genomic differential analysis in recent studies (*Oliva et al., 2020*; *Wang et al., 2018*; *Lopes-Ramos et al., 2020*; *Zhang et al., 2020*), which have the advantages of controlling all known (e.g., sex, age, race, etc.) and unknown variables (e.g., confounders and other batch effects, etc.). In each tissue, we fitted a linear regression model including age, sex, and age-by-sex interactions as covariates, as well as other confounding factors. The following linear regression was used:

$$PSI_{ij} \sim \mu_i + \alpha SEX_j + \beta AGE_j + \gamma SEX_j \times AGE_j + \sum_{k=1}^{N} \delta_k SV_{kj} + \varepsilon_{ij} \tag{4}$$

$$TPM_{ij} \sim \mu_i + \alpha SEX_j + \beta AGE_j + \gamma SEX_j \times AGE_j + \sum_{k=1}^{N} \delta_k SV_{kj} + \varepsilon_{ij} \tag{5}$$

where $PSI_{ij}$ is the normalized PSI value for AS event $i$ in sample $j$; $\alpha, \beta, \gamma$ are the coefficients of sex, age, and age-by-sex interactions, respectively; $\mu_i$ is the regression intercept; $\varepsilon_{ij}$ is the error term; and $k$ is the number of SVs in $N$ total SVs for sample $j$ with the coefficients $\delta_k$. SVs are estimated using SVA (surrogate variable analysis) (*Leek et al., 2012*). For differential GE analysis, we used the same linear regression model for analyzing differential GE by replacing PSI with TPM in the formula (*Equation 5*). To deal with the different orders of magnitude between PSI and TPM values, we normalized the PSI/TPM of each gene/AS event across samples using a scaling method to normalize the distance from the mean PSI/TPM across all samples.

$$norm\left(Y_{ij}\right) = \frac{Y_{ij} - mean\left(Y_j\right)}{AveDis\left(Y_j\right)}; AveDis\left(Y_j\right) = \frac{\sum_{i=1}^{n} |Y_{ij} - mean\left(Y_j\right)|}{n}$$

Here, $Y_{ij}$ indicates the PSI or TPM of the gene/AS event $i$ in sample $j$, and the $AveDis\left(Y_j\right)$ indicates the average distance from the mean PSI/TPM across all samples.

We defined sex- or age-differential genes with the cutoffs of p-value <0.05 for the *a* or *b* coefficients, and fold change >1.5. Sex- or age-differential AS events were defined with the cutoffs of p-value <0.05 for the a or b coefficients, and |ΔPSI| >0.05. We performed the differential analysis in each gene or AS event independently. The $\gamma$ significantly larger or smaller than 0 indicates significant

age-by-sex interaction, which suggests that the values of TPM and PSI are not only determined by sex or age independently but also affected by the age at a specific sex or sex at a specific age.

To evaluate whether the genetic effects could be controlled by SVs, we calculated the correlation between SVs and donors' ethnicity using Point Biserial Correlation which measures the association between continuous variables and dichotomous variables. Also, the Spearman's correlation was used to calculate the correlation between SVs and the top 5 principal components as judged by WGS data. The ethnicity information was extracted from phenotype data. The results of principal components calculated by the WGS dataset were downloaded from the GTEx portal.

Permutation analysis was used to address the false discovery issue of this linear regression model in the differential analysis. We first generated the permuted data by randomly shuffling the group labels (sex or age) with the same sample sizes as the original group labels. The differential genes/AS events (DEGs/DASs) were identified based on each generated permuted phenotype data. This process was performed 1,000 times and we used two strategies to examine the robustness of our cutoffs. We first defined the fractions of original DEGs/DASs which are also identified in each permutation iteration. Meanwhile, the FDR is defined as how many permutations incorrectly identified the specific DEG/DAS across 1,000 permutations.

## Evaluation of the effect of the sample sizes in differential GE and AS analysis

The effects of GTEx sample sizes on the numbers of differential genes/AS events were evaluated by a linear regression model by the following formula:

$$N_i \sim \mu + \alpha Size_i + \beta Detected_i + \varepsilon \tag{6}$$

where $N$ is the number of sex- or age-differential genes/AS events in tissue $i$; is the number of selected samples; $Detected$ is the detected numbers of genes/AS events after filtering in preprocessing procedures; $\alpha, \beta$ are the coefficients of the sample sizes and numbers of detected genes/AS events, respectively; $\mu$ is the regression intercept; and $\varepsilon$ is the error term.

## Identification of sex-stratified age-associated genes/AS events

To identify the genes/AS events affected by age in separate sexes, we also conducted differential analysis by fitting the following linear regression model in females and males, respectively:

$$Y_{ij} \sim \beta_0 + \beta_{F/M}AGE_j + \sum_{k=1}^{N} \delta_k SV_{kj} + \varepsilon_{ij} \tag{7}$$

where $Y_{ij}$ is the normalized TPM or PSI values of gene/AS event $i$ in sample $j$; $\beta_{F/M}$ is the coefficient of age in females or males; $\beta_0$ is the regression intercept; $\varepsilon_{ij}$ is the error term; and $k$ is the number of SVs in $N$ total SVs for sample $j$ with the coefficients $\delta_k$ evaluated in each sex. Age-associated genes or AS events were defined according to a significant age effect in different sexes ($\beta_M$, $\beta_F$, with cutoffs of $p < 0.05$; GE fold change >1.5 or AS |ΔPSI| >0.05). The biases in different sexes also reflected the functional interactions between sex and age. This approach not only could indicate whether there is a sex-by-age interaction, but also define the direction of sex-by-age interaction, that is, how females/males affect GE/AS during aging differently. In addition, ΔPSI and Fold change have also been used for filtering, which are the important thresholds to evaluate the degree of changes. Thus, we further defined the female- or male-specific age-associated genes/AS events using this sex-stratified analysis and classified these age-associated genes/AS events into sex-biased age-associated genes/AS events (i.e., age-associated genes or AS events specific to one sex) or sex-ubiquitous age-associated genes/AS events (i.e., age-associated genes or AS events in both sexes).

## Functional enrichment and disease ontology analysis

Potential functions of sBASEs were expounded based on C5 collections (ontology gene sets) in Molecular Signatures Database (MSigDB) and Disease Ontology. We carried out these processes using Bioconductor R packages *msigdbr* (**Dolgalev, 2020**), *clusterProfiler* (**Yu et al., 2012**), and *DOSE* (**Yu et al., 2015b**), with the significance threshold p-value <0.05. The GO terms with the significances specific to only one sex per tissue were selected for downstream analysis. In the enrichment heatmap of brain regions (**Figure 4A**), we plotted the GO terms with significance in at least 3 brain regions.

Because the enrichment heatmap across all tissues is unrealistically large, we plotted the terms within at least 15 tissues and the tissues/sex with significance in at least 9 significant GO terms (*Figure 3—figure supplement 4*). The entire results of the enrichment analysis are shown in *Supplementary file 4*.

## Processing of Alzheimer's disease datasets and construction of AD prediction model

The AD samples of human prefrontal cortex were selected from GEO datasets (ID: GSE174367) containing 375 samples of single-nucleus RNA-seq, single-nucleus ATAC-seq, and bulk RNA-seq. We filtered the datasets by selecting the RNA-seq samples from the Frontal Cortex brain region (Library Source: 'TRANSCRIPTOMIC'; Brain region: 'FC'). Next, we selected the samples with the isolation approach 'total RNA isolation'. Due to the sequencing depth on the splicing junction, we filtered out the samples lower than 1 Gb Bytes (SRR14514282, SRR14514349, SRR14514308, and SRR14514275). The remaining 91 samples were selected for GE and AS quantification, including 45 AD samples and 46 control samples from 47 male and 44 female donors (Female:Male ≈ 1:1; AD:Control ≈ 1:1) with similar age distribution between AD and control groups. Quality control and adaptor trimming were performed using FastQC (*Andrews, 2010*) (version 0.11.9) and Trim Galore (version 0.6.6). Bowtie2 (*Langmead and Salzberg, 2012*) (version 2.4.2) was used to remove reads mapping to ribosomal RNA. Then, clean reads were mapped to human reference (GRCh38) using STAR (*Dobin et al., 2013*) (version 2.7.6). We also used pre-mapped RNA-seq data from ROSMAP containing the RNA-seq samples from the bulk sequencing of DLPFC. We further filtered the data to match the number of both sexes and the age distribution between AD and control groups, resulting in the 215 AD and 56 control samples. The quantifications of GE and AS were performed by Paean (*Li et al., 2018a*).

To identify AD-related genes/AS events, we fitted a linear regression model with AD diagnosis (categorical outcomes) and SVs calculated by SVA (*Leek et al., 2012*). AD-related genes/AS events were defined whose coefficients of the 'diagnosis' term significantly deviated to zero. To compare the correlation coefficients that evaluated the correlation between AD- and age-associated AS changes in males and females independently, we performed Fisher's z-transformation on Pearson correlation coefficients using R packages *cocor* (*Diedenhofen and Musch, 2015*).

To capture the key age-associated splicing changes in males and females that were particularly involved in AD pathogenesis, we focused on the frontal cortex that was relevant to AD and further used the sBASEs for AD prediction. We randomly divided 90% samples for training sets and 10% for test sets and repeated this for 100 iterations. In each iteration, we performed a fivefold cross-validation with the feature selection approach called Recursive Feature Elimination via R packages *caret* (*Kuhn and Johnson, 2013*). Model performance was evaluated by the area under the curve. Random forest and support vector machine with a linear kernel were used for classification (*Figure 4D* and *Figure 4—figure supplement 2D*). To evaluate the final selected features that crucially contributed to AD prediction, we calculated the importance of each feature in each iteration (i.e., MDA) and then ranked the features by the averaged MDAs across 100 iterations. As negative controls, the sex-stratified model predicted AD patients in females and males separately using randomly selected AS events, and the merge-sexes model predicted AD patients in all samples regardless of sex labels using sBASEs.

## Transcriptome-wide association analysis with AD pathology

We used a linear regression model to identify the associations between AS events and AD pathology. The pathology includes tangles and plaque stages with continuous outcomes. Age was included as one of the covariates. This association analysis was performed for each sex.

## Construction of AS regulatory network

Splicing-related genes were extracted from the GO_RNA_SPLICING term based on MsigDB (c5.all. v7.4.symbols.gmt). Networks between age-associated SFs and sBASEs in each sex were constructed with the following three steps.

First, we examined the significant correlations between the expression of SFs and splicing of AS events during aging. We conducted Spearman's correlation test between TPMs of SFs and PSIs of AS events during aging using the threshold p-value <0.05.

Second, we chose the AS events significantly perturbed by RBP knockdown for downstream analysis. RBP shRNA RNA-seq bam files of 200 RBPs in HepG2 and K562 were downloaded from ENCODE Consortium. The TPM and PSI values were also quantified by Paean (*Li et al., 2018a*). Differential analysis between control and shRNA groups was performed by a linear regression model:

$$PSI_{ij} \sim \mu_i + \beta_G Group_j + \varepsilon_{ij}, \tag{8}$$

where $Group_j$ represents a categorical variable labeled by 'Control' or 'shRNA'. AS events significantly perturbed by RBP knockdown were defined by the estimated coefficient $\beta_G$ significantly deviating from zero.

Thirdly, we identified the AS events with SF-binding motifs around the alternative splice sites (SS). We used DeepBind (*Alipanahi et al., 2015*) to predict the specific binding scores of RBPs within the upstream to downstream region adjacent to the alternative SS. BEDTools getfasta (*Quinlan and Hall, 2010*) extracted sequences from specific genomic regions in a .bed6 file. We first subdivided the sequences into multiple subsequences (window = 40 nt; step = 1 nt) and calculated the binding scores with the parameter 'average' of each subsequence. To identify whether there were binding signals in the upstream to downstream 300-nt region adjacent to the alternative splice sites, we rearranged the binding scores in each region into small bins (width = 20) and tested the median differences between the bins with maximum and minimum averaged binding scores by Wilcoxon rank-sum test. The AS events whose maximum binding scores >1 and p-value <0.05 were reserved. Taken together, we required the regulatory connections between the RBPs and AS events that pass all the individual criteria as judged by all three types of experiments with different techniques. The networks between age-associated SFs and sBASEs were constructed and figured by Cytoscape (*Shannon et al., 2003*).

## Regulation of hormone receptors to the target genes

To examine the estrogen regulation using the RNA-seq dataset, we downloaded the RNA-seq datasets from MCF-7 cell line stimulated by estrogen receptor agonist (propyl pyrazole triol) in three biological replicates (GEO ID: GSE203058) (*Cao et al., 2024*). Additional datasets (ID: GSE89888) are downloaded to confirm our findings in engineered MCF-7 cell lines (*Bahreini et al., 2017*) with wild-type and two constitutively active mutants (Y537S and D538G) of ESR1. These cells were treated with 1 nM estradiol (E2) or vehicle control (veh) for 24 hr before collecting the total RNAs and subjected to transcriptome profiling by RNA-seq. We analyzed these data using the same pipelines in earlier Alzheimer's disease datasets (*Figure 4*). The GE changes under estrogen treatments were evaluated in a linear regression model using the p-values of the coefficients deviated from zero. To confirm the expression changes of SFs in other cells or tissues, we downloaded the FPKM table from GSE86609 that analyzed the ER knockout mice with estrogen treatments in mouse brain arcuate nucleus (ARCs) (*Yang et al., 2017*).

To further identify the regulation of the estrogen and androgen using ChIP-seq dataset (*Figures 5E and 6H*), we evaluated the binding scores of ESR1 and AR to the promoter regions of the target genes (i.e., ±1 kb of the transcription start sites, TSS). The binding score of each target gene (i.e., $-10 \times \log_{10}$[MACS Q-value] from ChIP-seq analysis) was calculated by the ChIP-Atlas (*Oki et al., 2018*), an integrative database of public ChIP-seq data. The ratios in *Figure 5E* were calculated by the binding scores of each SF divided by the median binding score of age-associated SFs common in both sexes. Additionally, the ESR1 ChIP-seq peak files in WT mice treated by estrogen were also downloaded from the Gene Expression Omnibus (ID: GSE3645) (*Hewitt et al., 2014*). The peaks were annotated using ChIPSeeker (*Yu et al., 2015a*).

## Time series and breakpoint analysis

ARIMA model is one of the linear models for forecasting chronological trends. For non-stationary data, the *auto.arima*() function from the R package *forecast* conducts a stepwise procedure to search for the best model using the smallest Akaike Information Criterion (AIC). This procedure is a variation of the Hyndman–Khandakar algorithm (*Brockwell et al., 2016*).

We first corrected the TPM/PSI values for each gene/AS event by removing the unknown confounding factors except for the sex and age effect via R package *limma* and averaged the values of the samples with the same sex and age. Then, we applied ARIMA models to figure out genes with potential chronological trends during aging in each sex. In brief, the input data was first transformed

into *z*-ordered objects (R package *zoo*) and the best model was stepwise optimized by AIC (*auto. arima* function from R package *Forecast*). We filtered out the genes/AS events with the significant trend for further analysis by non-seasonal difference order higher than zero ($D > 0$) and the sum of AR or MA coefficient higher than zero (AR +MA > 0).

Furthermore, we calculated the rate of change at each age point and estimated the breakpoints during aging. The rate of change at each age $i$ was evaluated by the differences between bins in $(i - w)\, i$ and $i\,(i + w)$ in the following steps. $w$ indicated a series of window spans from 5 to 15 years old. The p-values were first tested by MANOVA of three PCs after reducing dimensionality in the PCA process and then smoothed by LOESS regression with the smoothing bandwidth in the range from 0.25 to 0.75 (with interval 0.05). The p-values of each window span were integrated by average $-\log_{10}$ transformed p-values and smoothed by LOESS regressions with bandwidth = 0.5. The final rate of changes at each age point was defined by the averaged values through all window spans calculated in the previous step. Moreover, we determined the breakpoints as the age points at the global maximums, as well as the local maximums whose distance to the nearest minimum relative to the global maximum was more than 10%. This model was conducted based on the approach from a previously published study (*Márquez et al., 2020*; https://github.com/UcarLab/SexDimorphismNatureCommunications, *Mellert and Ucar, 2019*).

## Identification of AMGs

We performed the following steps to identify the AMGs that serve as the key contributors to molecular aging (*Figure 6*). We randomly trimmed 20% of chronological genes 200 times and repeated breakpoint analysis in each iteration. The Wilcoxon signed-rank test was used to compare the changing rate after removal with that calculated before removal. Using a p-value <0.05, we obtained the set of genes that could significantly alter the aging rate. The AMGs were defined as the set of genes that were significantly enriched from the 200 random samplings (as judged by the Fisher exact test using a p-value <0.05). The tissues with more than 10 chronological genes were selected for AMG identification process. Functional enrichment of sex-biased AMGs was based on MSigDB, and the network between GO terms and genes was constructed by Cytoscape (*Shannon et al., 2003*). GE patterns during aging were shown by the R package *pheatmap*.

Our results observed clusters of AMGs common in both sexes in whole blood tissue with sex-specific changing patterns (i.e., some genes increased first at young ages and then decreased during male aging, whereas these genes monotonically decreased during female aging). To identify AMGs with this expression pattern, we divided the age points into 9 age windows (e.g., 20–25, 26–30, 31–35, etc.) and averaged the ARIMA-fitted expression levels of the samples within each age window. The AMGs increased at young ages were defined by the mean of fitted expression levels at 31–40 larger than at 20–30.

## Acknowledgements

The authors want to thank Dr. Yue Hu and Jiefu Li in Wang Lab for their discussions and comments. We also thank Prof. Guoqing Zhang in SINH for his assistance with public data access and Ruijie Yao for helping download RBP shRNA-seq bam files from ENCODE Consortium and providing Paean scripts for multi-processing on the CPU–GPU platform. This work is supported by the Strategic Priority Research Program of CAS (XDB38040100), the Natural Science Foundation of China (91940303, 32030064, and 31730110), and the National Key Research and Development Program of China (2018YFA0107602) to ZW. ZW is supported by the type A CAS Pioneer 100-Talent program. This work is also supported by the National Key Research and Development Program of China to XL (2021YFA0805200 and 2019YFC1315804), and the National Natural Science Foundation of China to XL (31970554).

## Additional information

### Funding

| Funder | Grant reference number | Author |
|---|---|---|
| Strategic Priority Research Program of CAS | XDB38040100 | Zefeng Wang |
| National Natural Science Foundation of China | 91940303 | Zefeng Wang |
| National Natural Science Foundation of China | 32030064 | Zefeng Wang |
| National Natural Science Foundation of China | 31730110 | Zefeng Wang |
| National Key Research and Development Program of China | 2018YFA0107602 | Zefeng Wang |
| National Key Research and Development Program of China | 2021YFA0805200 | Xin Li |
| National Key Research and Development Program of China | 2019YFC1315804 | Xin Li |
| National Natural Science Foundation of China | 31970554 | Xin Li |

The funders had no role in study design, data collection, and interpretation, or the decision to submit the work for publication.

### Author contributions

Siqi Wang, Conceptualization, Data curation, Software, Formal analysis, Supervision, Validation, Visualization, Methodology, Writing – original draft, Project administration, Writing – review and editing; Danyue Dong, Resources, Data curation, Methodology, Writing – review and editing; Xin Li, Resources, Data curation, Supervision, Methodology, Writing – review and editing; Zefeng Wang, Conceptualization, Supervision, Funding acquisition, Investigation, Methodology, Writing – original draft, Project administration, Writing – review and editing

### Author ORCIDs

Siqi Wang ⬤ https://orcid.org/0000-0002-1491-2962
Zefeng Wang ⬤ https://orcid.org/0000-0002-6605-3637

Reviewer #2 (Public review): https://doi.org/10.7554/eLife.102449.3.sa1
Reviewer #3 (Public review): https://doi.org/10.7554/eLife.102449.3.sa2
Author response https://doi.org/10.7554/eLife.102449.3.sa3

---

## Additional files

### Supplementary files

MDAR checklist

Supplementary file 1. Summary for sample sizes of each group in multiple tissues.

Supplementary file 2. Principal component-based signal-to-variation ratio (pcSVR) values and corresponding permutation p-values between different age or sex groups as judged by gene expression (GE) and alternative splicing (AS).

Supplementary file 3. Lists of sex-stratified age-associated genes and alternative splicing (AS) events in 35 human tissues and brain regions.

Supplementary file 4. p-values of GO analysis for sex-biased age-associated alternative splicing events (sBASEs) across multiple tissues.

Supplementary file 5. AS regulatory networks between sex-biased age-associated alternative splicing events (sBASEs) and age-associated splicing factors in four functional brain regions.

Supplementary file 6. Aging-modulated genes (AMGs) and enriched p-values across multiple tissues.

## Data availability

The current manuscript is a computational study, so no data have been generated for this manuscript. GTEx data were obtained through dbGaP (accession number phs000424.v8.p2). The source RNA-seq data (*.fastq) of AD and control samples were obtained from Gene Expression Omnibus (accession number GSE174367). The RNA-seq and ChIP-seq data with estrogen treatment in ER mutant and wild-type cells were downloaded from Gene Expression Omnibus (accession number GSE203058, GSE89888, GSE86609, and GSE36455). Pre-mapped RNA-seq data (*.bam) of AD and control samples were obtained from ROSMAP (accession number syn8540863). shRNA-seq data of 200 RBPs in HepG2 and K562 cell lines were downloaded from ENCODE consortium (https://www.encodeproject.org). Molecular Signatures Database, (http://software.broadinstitute.org/gsea/msigdb/collections.jsp). The source code is available at GitHub (copy archived at *Wang, 2025*) and from Zenodo.

The following previously published datasets were used:

| Author(s) | Year | Dataset title | Dataset URL | Database and Identifier |
|---|---|---|---|---|
| GTEx Consortium | 2020 | Common Fund (CF) Genotype-Tissue Expression Project (GTEx) | https://www.ncbi.nlm.nih.gov/projects/gap/cgi-bin/study.cgi?study_id=phs000424.v8.p2 | dbGaP, phs000424.v8.p2 |
| Bennett DA, Buchman AS, Boyle PA, Barnes LL, Wilson RS, Schneider JA | 2019 | The Religious Orders Study and Memory and Aging Project (ROSMAP) Study | https://adknowledgeportal.synapse.org/Explore/Studies/DetailsPage/StudyDetails?Study=syn3219045 | AD Knowledge Portal, syn8540863 |
| Morabito S, Miyoshi E, Swarup V | 2021 | Single-nucleus chromatin accessibility and transcriptomic characterization of Alzheimer's Disease | https://www.ncbi.nlm.nih.gov/geo/query/acc.cgi?acc=GSE174367 | NCBI Gene Expression Omnibus, GSE174367 |
| Cao L, Ruan Z, Yang Y, Zhang N, Gao C, Cai C, Zhang J, Hu M, Shu H | 2022 | Estrogen receptor α-mediated signaling inhibits type I interferon response to promote breast cancer | https://www.ncbi.nlm.nih.gov/geo/query/acc.cgi?acc=GSE203058 | NCBI Gene Expression Omnibus, GSE203058 |
| Bahreini A, Oesterreich S | 2017 | RNA-seq analysis of ESR1 mutations in T47D and MCF7 cell lines | https://www.ncbi.nlm.nih.gov/geo/query/acc.cgi?acc=GSE89888 | NCBI Gene Expression Omnibus, GSE89888 |
| Yang JA, Stires H, Belden W, Roepke TA | 2017 | The Arcuate Estrogenome: Estrogen Response Element-dependent and -independent Signaling of ERα | https://www.ncbi.nlm.nih.gov/geo/query/acc.cgi?acc=GSE86609 | NCBI Gene Expression Omnibus, GSE86609 |
| Hewitt SC, Li L, Grimm SA, Chen Y, Liu L, Li Y, Bushel PR, Fargo D, Korach KS | 2012 | ERα and PolII ChIP seq from Mouse Uterus | https://www.ncbi.nlm.nih.gov/geo/query/acc.cgi?acc=GSE36455 | NCBI Gene Expression Omnibus, GSE36455 |

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
