## [Editor Report · eLife Assessment]

In this study Wang et. al. mined publicly available RNA-seq data from The Genotype-Tissue Expression (GTEx) database spanning multiple tissues to ask the question of how transcriptomes are changed with age and in both sexes. The authors provide **solid** evidence reporting widespread gene expression changes and alternative splicing events that vary in an age- and sex-dependent manner. An **important** finding is that many of these changes coincide with the time sex hormones begin to decline; additionally, the rate of aging is faster in males than in females.

---

## [Referee Report · Reviewer #2 (Public review)]

Summary:

In this manuscript, Wang et al analyze ~17,000 transcriptomes from 35 human tissues from the GTEx database and address transcriptomic variations due to age and sex. They identified both gene expression changes as well as alternative splicing events that differ among sexes. Using breakpoint analysis, the authors find sex dimorphic shifts begin with declining sex hormone levels with males being affected more than females. This is an important pan-tissue transcriptomic study exploring age and sex-dependent changes although not the first one.

Strengths:

(1) The authors use sophisticated modeling and statistics for differential, correlational and predictive analysis.

(2) The authors consider important variables such as genetic background, ethnicity, sampling bias, sample sizes, detected genes etc.

(3) This is likely the first study to evaluate alternative splicing changes with age and sex at a pan-tissue scale.

(4) Sex dimorphism with age is an important topic and is thoroughly analyzed in this study.

---

## [Referee Report · Reviewer #3 (Public review)]

Summary:

In this study, Wang et al utilized the available GTEx data to compile a comprehensive analysis that attempt to reveal aging-related sex-dimorphic gene expression as well as alternative splicing changes in human. The key conclusions based upon their analysis are that (1) extensive sex-dimorphisms during aging with distinct patterns of change in gene expression and alternative splicing (AS), and (2) the male-biased age-associated AS events have a stronger association with Alzheimer's disease, and (3) the females-biased events are often regulated by several sex-biased splicing factors that may be controlled by estrogen receptors. They further performed break-point analysis and reveal in males there are two main breakpoints around ages 35 and 50, while in female only one breakpoint at 45.

Strengths:

This study sets an ambitious goal, leveraging the extensive GTEx dataset to investigate aging-related, sex-dimorphic gene expression and alternative splicing changes in humans. The research addresses a significant question, as our understanding of sex-dimorphic gene expression in the context of human aging is still in its early stages. Advancing our knowledge of these molecular changes is vital for identifying therapeutic targets for age-related diseases and extending human healthspan. The study is highly comprehensive, and the authors are commendable for their attempted thorough analysis of both gene expression and alternative splicing-an area often overlooked in similar studies.

---

## [Author Response]

The following is the authors’ response to the original reviews

**Public Reviews:**

**Reviewer #1 (Public review):**
Summary:Wang et al. investigate sexual dimorphic changes in the transcriptome of aged humans. This study relies upon analysis of the Genotype-Tissue Expression dataset that includes 54 tissues from human donors. The authors investigate 17,000 transcriptomes from 35 tissues to investigate the effect of age and sex on transcriptomic variation, including the analysis of alternative splicing. Alternative splicing is becoming more appreciated as an influence in the aging process, but how it is affected by sexual dimorphism is still largely unclear. The authors investigated multiple tissues but ended up distilling brain tissue down to four separate regions: decision, hormone, memory, and movement. Building upon prior work, the authors used an analysis method called principal component-based signal-to-variation ratio (pcSVR) to quantify differences between sex or age by considering data dispersion. This method also considers differentially expressed genes and alternative splicing events.Strengths:(1) The authors investigate sexual dimorphism on gene expression and alternative splicing events with age in multiple tissues from a large publicly available data set that allows for reanalysis.(2) Furthermore, the authors take into account the ethnic background of donors. Identification of agingmodulating genes could be useful for the reanalysis of prior data sets.Weaknesses:The models built off of the GTEx dataset should be tested in another data set (ex. Alzheimer's disease) where there are functional changes that can be correlated. Gene-length-dependent transcription decline, which occurs with age and disease, should also be investigated in this data set for potential sexual dimorphism.

We appreciate the reviewer’s constructive feedback and acknowledgment of the strengths of our study. The detailed results are included in the ‘Recommendations for the authors’ from the editorial office. Below we summarize our feedback that address the concerns of this reviewer:

(1) Independent Alzheimer’s disease (AD) datasets:

We acknowledge the importance of validating our models beyond GTEx to assess their generalizability aging to Alzheimer’s disease. While GTEx provides valuable transcriptomic data across multiple tissues, it lacks direct functional assessments linked to disease states. We have already analyzed RNA-seq data from ROSMAP and GEO in Figure 4, focusing on sex-biased gene expression and splicing changes between aging and AD. The results showed a male-biased association with Alzheimer’s disease at AS resolution, indicating that the AS changes during aging could contribute more to AD in males than females. We added a highlight to this analysis in the manuscript (Pages 6-7).

(2) Sexual dimorphism in Gene-Length-Dependent Transcription Decline (GLTD)

We appreciate the reviewer’s suggestion to explore gene-length-dependent transcription decline (GLTD), which has been implicated in both aging and disease. As the reviewer suggested, our analysis revealed that GLTD exhibits sex-biased patterns in different tissues, aligning with recent literature on sex-dimorphic transcriptional aging. Our findings also revealed that longer genes with greater transcriptional decline are enriched in AD-related pathways. We have incorporated this new analysis in the ‘Recommendations for the authors’ in Author response image 5-6 and expanded the discussion of the biological relevance.

**Reviewer #2 (Public review):**
Summary:In this manuscript, Wang et al analyze ~17,000 transcriptomes from 35 human tissues from the GTEx database and address transcriptomic variations due to age and sex. They identified both gene expression changes as well as alternative splicing events that differ among sexes. Using breakpoint analysis, the authors find sex dimorphic shifts begin with declining sex hormone levels with males being affected more than females. This is an important pan-tissue transcriptomic study exploring age and sex-dependent changes although not the first one.Strengths:(1) The authors use sophisticated modeling and statistics for differential, correlational, and predictive analysis.(2) The authors consider important variables such as genetic background, ethnicity, sampling bias, sample sizes, detected genes, etc.(3) This is likely the first study to evaluate alternative splicing changes with age and sex at a pan-tissue scale.(4) Sex dimorphism with age is an important topic and is thoroughly analyzed in this study. Weaknesses:(1) The findings have not been independently validated in a separate cohort or through experiments. Only selective splicing factor regulation has been verified in other studies.(2) It seems the authors have not considered PMI or manner of death as a variable in their analysis.(3) The manuscript is very dense and sometimes difficult to follow due to many different types of analyses and correlations.(4) Short-read data can detect and quantify alternative splicing events with only moderate confidence and therefore the generalizability of these findings remains to be experimentally validated.

We appreciate the thorough review and thoughtful feedback. We have addressed the reviewer’s concerns and added clarification. The detailed results are included in Recommendations for the authors. Here are the summaries.

(1) Challenge of independent validation in separate cohorts

• The GTEx dataset includes the most comprehensive transcriptome resource for studying population-level differences in age and sex across tissues, particularly including large-scale brain samples. This provides a unique opportunity to analyze sex-dimorphic aging and the relevance of age-associated diseases. Several technical issues, including cell type heterogeneity, postmortem artifacts, as well as sequencing biases, lead to technical challenges in different cohorts.

• As the reviewer mentioned, we analyzed transcriptomic data from Shen et al. (2024) and compared them with GTEx results (Author response image 2). Limited overlap in differentially expressed genes again highlighted the challenges in cross-dataset validation due to the differences in cell composition and data processing (peripheral blood mononuclear cells (PBMCs) vs whole blood).

• Due to the limited human brain transcriptome data covering different age and sex groups, we found mouse hippocampus datasets from Mass spectrometry (MS), including young and old, as well as female and male groups. The results validated the expression of splicing factors in brain (Author response image 9). This cross-species consistency supports the robustness of our findings in human brain aging.

(2) Effects of Postmortem Interval, Manner of Death, and Time of Death

• We agree that the sample collections could introduce confounding effects. To address this, we calculated the correlations between the confounding factors with Postmortem Interval (PMI), Manner of Death (DTHMNNR), or Time of Death (DTHTIME and DTHSEASON). We observed strong correlations in some surrogate variables in most tissues, indicating that those factors could be well-regressed during our analysis (Recommendations for the authors, Figure S4 and R8).

• In addition, we re-evaluated our analyses while incorporating PMI as a covariate in our models. Our results align with our initial findings (Author response image 1), suggesting that age- and sex-dependent transcriptomic changes are not strongly confounded by PMI and confirming that our model has controlled PMI. These results are detailed in ‘Recommendations for the authors’ and included in Figure S4C-E with the description in text, Page 5.

(3) Readability of manuscript and flow of analyses

• In summary, our study first examined global alternative splicing (AS) and gene expression (GE) across all tissues before focusing on specific regions for deeper insights. To improve clarity, we have made the following revisions:

• Add clearer statements when transitioning between all-tissue and brain-specific analyses (Page 6-7).

• Modify the subtitle of Results to highlight all-tissue vs. brain analyses (Page 6).

• These refinements could enhance the manuscript’s structure, making the flow of analysis and conclusions more intuitive for readers.

(4) Limitations of short-read RNA-seq for splicing analysis

• Short-read RNA-seq provides only moderate confidence in detecting and quantifying full-length isoforms. However, its higher sequencing depth makes it more suitable for quantifying changes in alternative splicing (AS) events.

• Our analysis focused on splicing event-level quantification, applying stringent filters and using our GPU-based tool, which showed strong concordance with RT-PCR and other pipelines. Therefore, we also cited and included the updated Paean manuscript that benchmarks its performance in AS analysis.

**Reviewer #3 (Public review):**
Summary:In this study, Wang et al utilized the available GTEx data to compile a comprehensive analysis that attempt to reveal aging-related sex-dimorphic gene expression as well as alternative splicing changes in humans.The key conclusions based on their analysis are that.(1) extensive sex-dimorphisms during aging with distinct patterns of change in gene expression and alternative splicing (AS), and(2) the male-biased age-associated AS events have a stronger association with Alzheimer's disease, and (3) the female-biased events are often regulated by several sex-biased splicing factors that may be controlled by estrogen receptors. They further performed break-point analysis and revealed that in males there are two main breakpoints around ages 35 and 50, while in females, there is only one breakpoint at 45.Strengths:This study sets an ambitious goal, leveraging the extensive GTEx dataset to investigate aging-related, sexdimorphic gene expression and alternative splicing changes in humans. The research addresses a significant question, as our understanding of sex-dimorphic gene expression in the context of human aging is still in its early stages. Advancing our knowledge of these molecular changes is vital for identifying therapeutic targets for age-related diseases and extending the human health span. The study is highly comprehensive, and the authors are commendable for their attempted thorough analysis of both gene expression and alternative splicing - an area often overlooked in similar studies.

We thank this reviewer for the insightful review and recognition of our study's significance. We agree with the reviewer on how to examine sex-dimorphic gene expression and alternative splicing in aging by using the GTEx dataset. This is indeed an essential aspect of developing potential therapeutic targets for agerelated diseases to promote human health span.

Weaknesses:Due to the inherent noise within the GTEx dataset - which includes numerous variables beyond aging and sex - there are significant technical concerns surrounding this study. Additionally, the lack of crossvalidation with independent, existing data raises questions about whether the observed gene expression changes genuinely reflect those associated with human aging. For instance, the break-point analysis in this study identifies two major breakpoints in males around ages 35 and 50, and one breakpoint in females at age 45; however, these findings contradict a recent multi-omics longitudinal study involving 108 participants aged 25 to 75 years, where breakpoint at 44 and 60 years was observed in both male and females (Shen et al, 2024). These issues cast doubt on the robustness of the study's conclusions. Specific concerns are outlined below:References:Ferreira PG, Muñoz-Aguirre M, Reverter F, Sá Godinho CP, Sousa A, Amadoz A, Sodaei R, Hidalgo MR, Pervouchine D, Carbonell-Caballero J et al (2018) The effects of death and post-mortem cold ischemia on human tissue transcriptomes. Nature Communications 9: 490.Shen X, Wang C, Zhou X, Zhou W, Hornburg D, Wu S, Snyder MP (2024) Nonlinear dynamics of multiomics profiles during human aging. Nature Aging.Wucher V, Sodaei R, Amador R, Irimia M, Guigó R (2023) Day-night and seasonal variation of human gene expression across tissues. PLOS Biology 21: e3001986.(1) The primary method used in this study is linear regression, incorporating age, sex, and age-by-sex interactions as covariates, alongside other confounding factors (such as ethnicity) as unknown variables. However, the analysis overlooks two critical known variables in the GTEx dataset: time of death (TOD) and postmortem interval (PMI). Both TOD and PMI are recorded for each sample and account for substantial variance in gene expression profiles. A recent study by Wucher et al.(Wucher et al, 2023) demonstrated the powerful impact of TOD on gene expression by using it to reconstruct human circadian and even circannual datasets. Similarly, Ferreira et al. (Ferreira et al, 2018) highlighted PMI's influence on gene expression patterns. Without properly adjusting for these two variables, confidence in the study's conclusions remains limited at best.

We appreciate the reviewer for raising this important point regarding the impact of post-mortem interval (PMI) and time of death (TOD) on gene expression, including the death seasons (DTHSEASON) and daytime (DTHTIME). To address this point, we carefully evaluated whether our linear model controlled for these factors as potential confounders.

Our results showed that PMI and TOD significantly correlated with the estimated covariates in most tissues, suggesting that their effects could be effectively regressed out using our model (Figure S4). As the reviewers and editors suggested, we have now included this correlation analysis in the updated Figure S4C-E and the text in the Results section, citing relevant literature [1,2] (Page 5).

**Author response image 1. sa3fig1:** The results of differential gene expression analysis with vs without the inclusion of PMI correction as a known covariate. The scatter plots show the correlations of significance levels (pvalues, left panel) and effect sizes (coefficients, right panel) of sex (A) and age (B). Whole-blood tissue is used as an example.

In addition, we did the differential analysis that incorporated PMI as a covariate in the regression models and re-evaluated the age- and sex-related transcriptomic changes. Using WholeBlood gene expression as an example, our revised analysis shows that the inclusion of PMI in the covariates has minimal impact on the significance levels and effects of sex and age (i.e., p-values and coefficients, respectively), indicating that our findings are robust using confounding factors (Author response image 1).

(2) To demonstrate that their analysis is robust and that the covariates TOD and PMI are otherwise negligible - the authors should cross-validate their findings with independent datasets to confirm that the identified gene expression changes are reproducible for some tissues. For instance, the recent study by Shen et al. (Shen et al., 2024) in Nature Aging offers an excellent dataset for cross-validation, particularly for blood samples. Comparing the GTEx-derived results with this longitudinal transcriptome dataset would enable verification of gene expression changes at both the individual gene and pathway levels. Without such validation, confidence in the study's conclusions remains limited.

We thank the reviewer for the insightful suggestion regarding cross-validation with independent datasets. We understand that validating findings across datasets is crucial for ensuring robustness. As the reviewers suggested, we see whether there are some shared findings in the GTEx data with the study by Shen et al. (2024) in Nature Aging. However, after performing comparisons with our GTEx results in whole blood tissue, we found that the overlaps of differentially expressed genes are limited (Fig. 3). In our results, we found a large proportion of age-associated genes in the GTEx data, whereas just 54 genes are age-associated from Shen et al.’s PBMC data. 3 in 7 genes are differentially expressed in both datasets (Fig. 3A). Additionally, we performed the functional enrichment analysis on the GTEx-specific age-associated genes.

We observed a strong enrichment in the biological pathways related to neutrophil functions and innate immune responses, which are specific to the cell compositions in whole blood rather than PBMC (Fig. 3B).

**Author response image 2. sa3fig2:** The comparison between the gene expression of whole blood tissue from GTEx and PBMCs from Shen et al. (A) The bar plot shows the number of age (left panel) or sex-associated (right panel) genes in the two datasets. The grey bars highlight the proportion of overlapped genes in both datasets. (B) The top 10 significantly enriched biological processes in the GTEx-specific age-associated genes. The color bar shows the number of age-associated genes in specific pathways.

These discrepancies highlighted the crucial factors in cross-dataset comparison:

• Cell compositions: GTEx used whole blood, which contains all blood components, including neutrophils and erythrocytes, whereas PBMCs contain lymphocytes and monocytes. Under the influence of granulocytes and red blood cells in whole blood, the gene expression profiles between these two datasets are different.

• Biological functions: Whole blood includes both innate and adaptive immune components; thus, aging-related gene expression changes in whole blood may include a broader systemic response than those in PBMCs. This difference in biological context contributes to the observed variation in the differentially expressed genes, as demonstrated by our functional enrichment analysis (Fig. 3B).

• Sequencing biases and data processing: The two datasets were generated using different RNAseq processing pipelines, including distinct normalization, batch correction, and quantification methodologies. These technical differences may introduce systematic variations that complicate direct cross-validation.

Due to these fundamental problems, a direct one-to-one validation between the two datasets is challenging. We understand the importance of independent dataset validation and appreciate the reviewer’s suggestion. However, future studies could be performed more precisely if comparable whole-blood-based datasets are available. In addition, GTEx data provides nearly thousands of samples in whole blood, which is a largescale, comprehensive, and clinically relevant dataset for studying aging-related changes, particularly in innate immunity and inflammation, which are not well captured in PBMCs.

(3) As a demonstration of the lack of such validation, in the Shen et al. study (Shen et al., 2024), breakpoints at 44 and 60 years were observed in both males and females, while this study identifies two major breakpoints in males around ages 35 and 50, and one breakpoint in females at age 45. What caused this discrepancy?

We thank the reviewer and the editors for both coming up with the non-linear multi-omic aging patterns observed by Shen et al. They observed two prominent crests around the ages of 45 and 60 from omics data.

Similarly, we also identified two breakpoints in our analysis, with some differences in specific age breakpoints. These could be the result of sample preparation methods and breakpoint definition. These responses are also included in the editor’s recommendations.

Definition of breakpoints vs crests:

• Crests represent age-related molecular changes at each time point across the human lifespan. They indicate the number of molecules that are differentially expressed during aging (q < 0.05), without considering individual expression levels.

• Our breakpoints, in contrast, are identified after filtering the chronological trends using the Autoregressive Integrated Moving Average (ARIMA) model. We calculated the rate of change at each age point using the smooth approach and sliding windows. Breakpoints are defined as local maxima where the distance to the nearest minimum, relative to the global maximum. We indeed found some local wide peaks around 60 in some tissues, shown in Figure S10, however, we excluded these due to our strict cutoffs to remove noise.

Differences and similarities between sequenced tissues:

• Whole-blood vs PBMC: In the GTEx RNA-seq data used in our study, whole blood samples from donors were sequenced, whereas their study used PBMCs. Whole blood contains all blood components, including red blood cells, platelets, granulocytes (e.g., neutrophils), lymphocytes, and monocytes, while PBMCs represent a subset of white blood cells, primarily consisting of lymphocytes (T cells, B cells, NK cells) and monocytes, excluding granulocytes and erythrocytes. As we mentioned in the previous responses, the gene expression changes observed in whole blood capture the contributions of neutrophils and other granulocytes, which are neglected in the PBMC profile (also shown in Figure S11C).

• For the shared tissues in two studies – skin, we looked at the non-linear changes during aging and found the same two breakpoints: 43 and 58.

Novelties in our study:

• Whole blood can serve as a readily accessible resource for testing age-related disease biomarkers without cell separation, making it more practical for clinical applications.

• Our analysis was performed on females and males, respectively. The main object of our analysis is to compare the differences in aging rates between sexes. Our results reveal clear sex-specific differences across multiple human tissues. Therefore, the identified breakpoints may differ when sex effects are not taken into account, highlighting the specificity of our analysis.

• Additionally, our breakpoints are integrated across multiple tissues. Our results showed that there is a large diversity of aging patterns in different tissues.

As the reviewers and editors suggested, we have added the following statements to clarify this distinction in the Discussion section: ‘Our analysis observed the non-linear aging patterns with two breakpoints, which is consistent with recent findings, with differences in specific age points due to sex differences as well as tissue diversities 3.’ (Page 14), and ‘These breakpoints could represent key junctures in the aging process that align with the non-linear patterns of aging and disease progression.’ (Page 15)

(4) Although the alternative splicing analysis is intriguing, the authors did not differentiate between splicing events that alter the protein-coding sequence and those that do not. Many splicing changes occurring in the 5' UTR and 3' UTR regions do not impact protein coding, so it is essential to filter these out and focus specifically on alternative splicing events that can modify protein-coding sequences.

The reviewer raises an important point. In our study, we included the AS events in protein-coding genes to gain a comprehensive understanding of sex-biased age-associated splicing. As the reviewer suggested, focusing on coding-sequence-altering events is particularly relevant to protein function. To address this, we performed an additional analysis to specifically annotate sBASEs occurring within the coding sequence (defeined as CDS-altering sBASEs) and reanalyzed their functional pathways and AD-associations (Author response image 3).

Our analysis revealed that most of the sBASEs are relevant to protein-coding sequences (CDS) across multiple tissues (Author response image 3A). We then confirmed our findings using CDS-altering sBASEs. We found that those sBASEs in brain regions were significantly enriched in pathways related to amyloid-beta formation and actin filament organization (Author response image 3B). Notably, male-biased sBASEs in decision-related brain regions were particularly associated with dendrite development and regulation of cell morphogenesis, highlighting the sex-specific roles of sBASEs in brain functions. Additionally, we performed a random forest classification using only CDS-altering sBASEs in AD datasets (Author response image 3C-D), again confirming the malebiased association between aging and AD.

Overall, we found that most of the identified sBASEs could modify protein-coding sequences, and our main conclusions remain consistent even after filtering out non-coding events.

Nevertheless, in addition to AS events that impact protein sequences, alternative splicing in untranslated regions (UTRs) also plays a critical regulatory role. Splicing events in the 5′ UTR can influence translation efficiency by modifying upstream open reading frames (uORFs) or RNA secondary structures, while splicing in the 3′UTR can affect mRNA stability, localization, and translation by altering microRNA binding sites and RNA-binding protein interactions. Given these functional implications, we believe that UTR-targeted AS events should also be considered to supplement the understanding of post-transcriptional gene regulation in future research.

**Author response image 3. sa3fig3:** The distribution and functional relevance of sBASEs with coding effects. (A) The number of sBASEs and CDS-altering sBASEs across multiple tissues. The deeper bars show the number of sBASEs whose alternative splice sites are located at protein-coding regions. (B) GO biological pathways in each sex and brain region. Heatmap shows the sex-specific pathways that are significantly enriched by CDS-altering sBASEs in more than 2 brain regions and sex. (C) Correlation between ADassociated and age-associated AS changes across the CDS-altering sBASEs that alter protein-coding sequences in females and males. (D) Performances of sex-stratified models predicted by CDS-altering sBASEs in 100 iterations using the random forest approach

(5) One of the study's main conclusions - that "male-biased age-associated AS events have a stronger association with Alzheimer's disease" - is not supported by the data presented in Figure 4A, which shows an association with "regulation of amyloid precursor formation" only in female, not male, alternative splicing genes. Additionally, the gene ontology term "Alzheimer's disease" is absent from the unbiased GO analysis in Figure S6. These discrepancies suggest that the focus on Alzheimer's disease may reflect selective data interpretation rather than results driven by an unbiased analysis.

We thank the reviewer for this point. In our functional analysis, we identified distinct biological processes enriched in female- and male-biased AS genes, such as the regulation of amyloid precursor formation in females and structural constituents of the cytoskeleton in males. However, Alzheimer’s disease (AD) is a complex neurodegenerative disorder with multiple pathological mechanisms beyond amyloid-beta (Aβ) formation, many of which are strongly age-related in both sexes. This complexity motivates us to explore novel relationships between splicing and AD in distinct sexes.

Although Figure 4A shows the enrichment of “regulation of amyloid precursor formation” in female-biased AS events, this does not contradict the broader enrichment of AD-related processes in male-biased AS events. Our disease ontology analysis supports this finding, as male-biased age-associated AS events are enriched in neurodegenerative diseases, including cognitive disorders. Additionally, we considered not only individual GO terms but also the disease-associated transcriptomic signatures from AD-related datasets, which collectively indicate a stronger association in males.

Regarding Figure S6 mentioned by the reviewer, the GO term “Alzheimer’s disease” is not explicitly listed in the heatmap because we filtered the pathways that are consistently enriched in multiple tissues. As noted in the figure legend, we only displayed sex-specific GO terms that were significant in at least 15 tissues. Then, since the brain is highly affected by age-related processes and neurological conditions show sex differences, the sex-biased AS events could help explain differential susceptibility to age-related cognitive decline and neurodegeneration. That’s why we chose the brain data for detailed analysis.

To improve clarity, we have revised the text to describe the purpose of our analysis in brain rather than other tissues (Page 6-7). We appreciate the reviewer’s feedback, and we will consider additional analyses to further explore the sex-biased AS as well as disease risk in other tissues.

(6) The experimental data presented in Figures 5E - I merely demonstrate that estrogen receptor regulates the expression of two splicing factors, SRSF1 and SRSF7, in an estradiol-dependent manner. However, this finding does not support the notion that this regulation actually contributes to sex-dimorphic alternative splicing changes during human aging. Notably, the authors do not provide evidence that SRSF1 and SRSF7 expression changes actually occur in a sex-dependent manner with human aging (in a manner similar to TIA1). As such, this experimental dataset is disconnected from the main focus of the study and does not substantiate the conclusions on sex-dimorphic splicing during human aging. The authors performed RNAseq in wild-type and ER mutant cells, and they should perform a comprehensive analysis of ER-dependent alternative splicing and compare the results with the GTEx data. It should be straightforward.

Thanks for the reviewer’s feedback. The main purpose of the analyses in Figures 5E-I was to explore which factors affect the sex-biased expression of splicing factors during aging and substantially regulate alternative splicing (AS). To address the reviewer’s concerns, we have included additional analysis and explained the challenge of linking estrogen receptor (ER)-regulated splicing factors to sex-dimorphic AS changes during human aging in specific human cell types.

• As suggested by the reviewer, we first examined the expression changes of SRSF1 and SRSF7 during aging in males and females, like TIA1 in decision-related brain regions (Fig. 5I).

• Secondly, the regulation is based on a highly complex regulatory network involving multiple splicing factors and cell heterogeneity. Due to these complexities, we did not overlap ER-dependent AS changes with sBASEs from GTEx datasets directly. As far as the reviewer is concerned, we supplemented the AS analysis in the GSE89888 dataset (Fig. 5H) and identified the estrogenregulated AS events mediated by ESR1. We found that ~6% (26/396) of female-specific ageassociated AS events were regulated by ESR1, of which 6 sBASEs can be regulated by femalebiased splicing factors. The low overlaps could be represented by the limited coverage of different RNA-seq datasets and cell types used across these analyses. Notably, the results indicated that only a fraction of AS could be directly accounted for by estrogen via ESR1, suggesting the complexity of transcriptional and splicing regulatory networks during aging.

• Meanwhile, we downloaded independent experimental datasets to discover the regulation by our candidate splicing factors. Due to SRSF1 is identified as a potential regulator of sex-biased splicing, we analyzed RNA-seq data with SRSF1 knock-down (KD) glioblastoma cell lines (U87MG and U251), a type of brain cancer formed from astrocytes that support nerve cells 4. As a result, we indeed found that some sBASEs are regulated by SRSF1 during aging through this experiment using brain cell lines (Author response image 4). Together, these results suggested that some of the SF-RNA regulatory relationships can be observed in another cellular system, further supporting our findings.

Due to the limitations of cell-based models and the complexity in the splicing regulatory network, it is challenging to directly validate aging regulation, particularly between different sexes, based on ER treatments in vivo. However, our findings still provide valuable mechanistic insights into ER-regulated splicing factors, implying their potential role in sex-biased aging.

**Author response image 4. sa3fig4:** SRSF1 regulations on specific sBASEs using SRSF1 knock-down RNA-seq data in GBM cells. Three examples are shown to be regulated during aging with significant changes between SRSF1 KD vs control in U251 and U87MG cell lines. The splicing diagrams are shown below.

**Recommendations for the authors:**

**Reviewer #1 (Recommendations for the authors):**
The authors found that alternative splicing was affected by both sex and age across many tissues, with gene expression differences affected by both parameters only present in some tissues. This trend was consistent when the effects of sex chromosomes were subtracted from the analysis. The effect of aging on differential gene expression and alternative splicing was more prevalent in male than female samples. For analysis purposes, young subjects were deemed to be anyone under 40, and old subjects were over 60 years old. The authors then investigated if specific genes or alternative splicing events were responsible for these effects. Some candidate genes or splicing events were identified but there was little overlap between tissues, suggesting no universal gene or event as a driver of aging. Surrogate variables like the ethnic backgrounds of donors were also investigated. Ultimately the authors found that alternative splicing events showed a stronger sexual dimorphic effect with age than did differential gene expression and that at least for the brain, alternative splicing changes showed a bias for Alzheimer's disease in male samples. This was highlighted by examples of exon skipping in SCL43A2 and FAM107A in males that were associated respectively with plaques and tangles.The authors go on to identify sexual dimorphic differences in splicing factors in particular brain regions during age. Finally, the authors performed analysis for aging-modulated genes, identifying nearly 1000 across the tissues, nearly 70% of which are sex-specific. Their work suggests that further analysis of these aging-modulated genes could be differentially modulating the transcriptome based on sex. The work is novel and interesting, especially investigating sexual dimorphism in alternative splicing. However, the work is still preliminary, and these assumptions need to be applied to other data sets beyond GTEx for validation as well as some other phenomena that need to be considered. I recommend major revisions to address the points below.(1) At the beginning of the results section, the authors state that the brain is stratified into four functional regions. It would be useful to explicitly state those four regions in the text at that point.

We agree that specifying these regions early in the text will improve clarity and provide the reader with a clear understanding of the analysis. As the reviewer’s suggestion, we revised the Results section (Page 3) to explicitly state the four functional brain regions as follows: ‘Due to data sparseness, the brain tissues were recombined into four functional regions (table S1), including hormone- or emotion-related region, movement-related region, memory-related region, and decision-related region (See Methods).’. This ensures that the regions are clearly defined before the subsequent analysis is presented.

(2) The manuscript becomes a bit confusing when the authors shift from all the tissues as a whole specifically to the brain and then back to the larger tissue set to make assumptions. This can be a bit confusing and should be better delineated.

We thank the reviewer and editor for the feedback regarding the transitions between the analysis of all tissues and the brain-specific analysis. In our study, we first conducted a broad analysis of alternative splicing (AS) and gene expression (GE) across all tissues. For the AS analyses, we did sBASEs analysis in all tissues and then focused on specific tissue (i.e., brain) whose splicing changes are functionally enriched with age-related diseases. For the GE analyses, we also analyzed the aging rate across tissues and identified the tissue-specific/shared patterns.

We agree that the shifts of the tissues for AS and GE may cause some confusion, and have made the following revisions to delineate why we focused on different tissues for distinct analyses:

• We have added clear statements to better delineate when we shift focus from the analysis of all tissues to the region-specific analysis and vice versa. For instance, in the Results section (Page 67), we include a transitional phrase: ‘Having established patterns across all tissues, we now turn to a more focused analysis to investigate tissue-specific alternative splicing changes.’

• To improve the overall structure, we have reorganized the Results section, adding distinct subheadings for the analysis of all tissues and the brain (Page 6), which should make the transition between these sections smoother and more intuitive for the reader.

We believe that these revisions will make the manuscript’s structure clearer and allow the reader to better follow the flow of the analysis and the subsequent conclusions.

(3) Gene-length-dependent transcription decline (GLTD) is another phenomenon that occurs with aging and is known to be associated with Alzheimer's disease [PMID38519330]. The authors should make some statement if this is present in their dataset and if any sexual dimorphism in tissues is present.

We thank the editors and reviewers for bringing up the possible connection of gene-length-dependent transcription decline (GLTD), which was reported to be associated with both aging and Alzheimer’s disease (AD). We appreciate the reviewer’s suggestion and have addressed whether GLTD is present in our dataset and whether any sex differences are observed in this context.

We evaluated GLTD using the correlation between gene length with age-associated changes (i.e., the coefficients of the ‘age’ term in the linear regression model) in GTEx data. We did observe strong evidence of GLTD, particularly in the brain, heart, muscle, pancreas, spleen, skin, muscle, etc (Author response image 5A). In brain, we performed the functional enrichment analysis on the genes with Foldchange > 2 and length > 10^5^ bp (Author response image 5B). We found that these extremely long genes are significantly relevant to synapse and neuron functions. These findings align with previous studies showing that GLTD can occur with aging in the tissues that are relevant to Alzheimer’s disease, cardiovascular diseases, and common failures of metabolism (e.g., diabetes) [5,6]. Additionally, it was not a ubiquitous phenomenon across all tissues. The correlations could be positive in tissues like adipose and artery. These findings suggested the GLTD could be varied and tissuespecific in its manifestation during aging.

**Author response image 5. sa3fig5:** (A) The correlation between gene length and age-associated changes across GTEx tissues in human samples. The correlation tests are evaluated using Spearman’s approach. The color bar indicates the -log10 transformed p-values in the correlation test. (B) The results of GO enrichment analysis using the genes with Foldchange > 2 and length > 10^5^ bp. The parent terms calculated by ‘rrvgo’ with a similarity threshold of 0.9 are shown.

Regarding sexual dimorphism, we conducted this analysis in females and males, respectively (Author response image 6). We found GLTD exists in both females and males in most tissues, such as brain, whole blood, muscle, etc, consistent with the previous results without considering the sex groups. Interestingly, we observed sexbiased patterns in certain tissues. In particular, the left ventricle, pancreas, and hippocampus showed notable male-biased patterns in the degree of transcriptional decline with gene length, whereas skin, liver, small intestine, and esophagus showed that in females. These findings suggest that GLTD could be relevant to aging and age-related diseases; the levels of expression and sexual dimorphism may vary depending on the tissue type. We hope this clarification addresses the reviewer’s concern and provides a more comprehensive understanding of the GLTD and sex differences observed in our dataset.

**Author response image 6. sa3fig6:** The correlation between gene length and age-associated changes across tissues in females and males, respectively. The correlation tests are evaluated using the Spearman’s approach. The red dots indicate the significant correlations in females, while the navy dots show those in males.

(4) Because the majority of this work has been performed in the GTEx dataset, applying this analysis to another publicly available dataset would be useful validation. For instance, the authors have interesting findings in the brain and correlations to Alzheimer's disease. Analysis of an existing RNAseq dataset from Alzheimer's disease patients and controls (with functional outcomes) would provide more evidence beyond the preliminary findings from GTEx.

We appreciate the reviewer’s suggestion on the validation of our findings by applying our analysis to independent RNA-seq datasets from Alzheimer’s disease patients.

• We have used two Alzheimer’s disease datasets, GEO and ROSMAP, to investigate the correlation between aging and Alzheimer’s disease (AD) and included these analyses in our study (Fig. 4B-C and Figure S8C).

• In the Results section (Page 7), we have presented the results of this validation, where we identified correlations between sex-biased aging-related splicing changes and AD-related changes. These findings support the conclusions from the GTEx dataset and further strengthen the relevance of our results to AD.

As suggested, we have updated the manuscript to more explicitly highlight this validation in the Discussion section (Page 12), noting: ‘We further validated our findings using Alzheimer’s disease dataset, ROSMAP, where we observed consistent correlations between aging-related splicing changes and Alzheimer’s disease-related changes, providing additional evidence for the robustness of our results.’

**Reviewer #2 (Recommendations for the authors):**
(1) In the text (Introduction and Discussion), the authors mention analyzing 54 tissues, the abstract states 35 tissues, Table S1 lists 48, and Figure 2A-B shows 33. Could the authors please clarify exactly how many tissues they used? I am also confused by the sample numbers in Table S1. For example: for adiposesubcutaneous tissue, the total number of females is listed as 218 but the sum of young and old females is only 110. Does this mean some samples were excluded? What is the exclusion criterion?

We thank the reviewers and editors for pointing out the discrepancies regarding the number of tissues analyzed and the sample numbers in Table S1. We appreciate the opportunity to clarify these points:

Number of tissues analyzed:

• We downloaded and analyzed 17,382 samples in 54 tissues from GTEx in total (31 tissues and 13 brain regions), as mentioned in the Results, Methods, and Discussion sections. Table S1 lists 48 tissues (31 tissues, 13 brain regions, and 4 merged brain regions), which include a refined classification of the tissues we analyzed, accounting for the variations in brain region categorization in the dataset.

• The discrepancy also arises from the different sample size cutoffs in specific analyses. For pcSVR analysis (Figure 2A-B), we did the subsampling for the permutation analysis for certain key findings, so we filtered a subset of 33 tissues (29 tissues and 4 merged brain regions), which included at least 3 samples in each age group in females or males.

• To resolve this, we have clarified the total number of tissues analyzed and aligned the numbers across the manuscript. In the revised manuscript, we now explicitly state in both the Abstract and Methods sections that 54 tissues were analyzed in the context of this study. We added a note in Methods to clarify that 35 tissues are 31 tissues and 4 merged brain regions (Page 16). In Figure 2A-B, we clarified that the 33 tissues are filtered due to the usage in this analysis (Page 17).

Sample numbers in Table S1:

• Regarding the sample sizes of age groups, the discrepancy occurred due to the classification of the age groups. We classify the samples into three: Young, Middle, and Old, as mentioned in the Results section (Page 4).

• Additionally, we excluded the sample sizes in 13 single brain regions. We aligned the total tissue number to 35 with our texts.

We hope this resolves the confusion regarding the number of tissues and the sample sizes used in the analysis. These clarifications have been incorporated into the revised manuscript to ensure consistency.

(2) Was post-mortem interval (PMI) or manner of death considered in the model? For example, traumatic death may have major consequences on gene expression. Similarly, a few tissues have low sample numbers, for example, kidney cortex and brain. The pooling of brain samples is explained and the kidney cortex is excluded, so why is it listed in Table S1?

Thank you for raising this important point regarding the potential impact of post-mortem interval (PMI) and manner of death (DTHMNNR) on gene expression. We carefully considered both factors as potential confounders in our analysis.

Specifically, to evaluate their impacts, we calculated the correlations between the coefficients of PMI or manner of death, with the confounding factors. Our results showed that PMI and DTHMNNR are significantly correlated with the covariates in most tissues, suggesting that their effects could be effectively regressed in our model (Figure S4). As we have mentioned in Figure S4 and Author response image 1, we conducted a differential analysis that incorporated PMI as a covariate in the regression models and re-evaluated the age- and sex-related transcriptomic changes to address this concern. The high correlations showed the minor effect size of PMI when including the covariates in the model. As suggested by the reviewers and editors, we have now included this correlation analysis in Figure S4C-E and updated the text in the results section (Page 5).

Additionally, as the responses above, Table S1 provides the general sample sizes of all GTEx tissues without filtering. We have modified the table to include a total of 35 tissues, including 31 non-brain tissues and 4 brain regions.

(3) It might be important to show a simple visual of cohort details such as age ranges, sexes, ethnicities, PMIs, etc.

To address this, we added summary figures to illustrate the distributions of key demographic variables, including age, sex, BMI, ethnicity, post-mortem intervals (PMIs), and manner of death (DTHMNNR) (Author response image 7 and Author response image 8). This will provide readers with a clearer overview of the dataset composition and potential covariates affecting the analysis.

**Author response image 7. sa3fig7:** Age (left panel), BMI (Body Mass Index) (middle panel), and PMI (Post-Mortem Interval) (right panel) distribution in GTEx v8 cohort.

**Author response image 8. sa3fig8:** Sex (left panel), ethnicity (middle panel), and manner of death (DTHMNNR) (right panel) distribution in GTEx v8 cohort.

(4) Since this study is highly correlative, it is impossible to determine if the findings hold true without an independent cohort validation or experimental validation. They used the ROSMAP cohort for AD samples, and some splicing factors regulation but the generalizability to the age and sex effects have not been independently tested.

The reviewer raises an important point regarding the independent validation of sex- and age-associated splicing changes associated with AD. We used GTEx primarily because it includes approximately 17,000 RNA-seq samples across multiple human tissues, making it the most comprehensive public resource for studying population-level differences in age and sex. In particular, its large-scale brain samples provide a unique opportunity to analyze transcriptomic changes in sex-dimorphic aging.

We understand the reviewer’s concern that our findings are mainly supported by correlative evidence, which could be affected by dataset-specific biases. However, there are several technical issues in crossvalidation with transcriptomes across different datasets, including limited comparability due to cell type heterogeneity, postmortem artifacts, and sequencing biases.

Specifically, GTEx data is bulk RNA-seq that does not capture cell-type-specific transcriptomic changes. Given the cellular complexity of the brain and other tissues, observed differences in gene expression and splicing may be influenced by shifts in cellular composition rather than intrinsic transcriptional regulation. For example, we compared our results from GTEx whole blood with the analysis using an external dataset from Peripheral Blood Mononuclear Cells (PBMCs) provided by Shen et al. (2024) [3] (Author response image 2). We observed limited overlap in differentially expressed genes between these datasets (probably because the whole blood contains diverse immune cell populations), highlighting the challenges in cross-dataset validation due to differences in tissue composition and sample processing.

Therefore, we applied surrogate variable analysis (SVA) to minimize technical and biological confounders. This approach helped reduce biases from genetic background to hidden batch effects, including postmortem artifacts, sequencing biases (Figure S4), and other covariates. This approach could help us identify whether sex-biased splicing events are biologically meaningful rather than technical artifacts.

In addition, to address the reviewer’s concern on the splicing factor regulation, we managed to find a dataset in decision-related brain regions. Due to the limitation of human brain data covering different age and sex groups, we used mouse hippocampus datasets, including young and old, as well as female and male groups [7]. The analysis of protein levels from MS data identified sex-biased age-associated splicing factors, including Srsf1 and Srsf7. We found that the changes are consistent with the findings from GTEx (Author response image 9), aligning with our sex-biased splicing factor expression during aging in the same region of the human brain. This cross-species consistency supports the robustness of our findings in human brain aging.

**Author response image 9. sa3fig9:** Protein levels of some male-specific splicing factors in human hippocampus quantified using MS data. The Y-axis shows the protein intensity. Different facets mean different sample batch sets. The yellow boxes indicate the protein levels in the young group, while the brown boxes indicate those in the old group.

In summary, despite the inherent limitations of RNA-seq studies in sex- and age-related transcriptomics, we have made our best efforts to address these concerns through comparisons with external datasets, statistical corrections, and validation using proteomic data. We appreciate the reviewer’s feedback and include additional discussion on these points (Page 13).

(5) Are AS predictions from short-read data accurate enough to make the predictions the authors report?

The reviewer is correct that the short-read sequencing has inherent limitations in reconstructing full-length isoforms. However, the higher sequencing depth for short reads makes it a better choice in quantifying the relative change of each AS event across different conditions. As a result, short-read data are extensively used in the splicing field to quantitatively measure the AS changes. For this reason, we focused on the levels of alternative splicing events, rather than the quantification of full-length isoforms. We used a series of stringent filters in our analyses to increase the reliability of our results.

Specifically, we filtered the read counts of the junction read counts (JC) of most differential AS events that were higher than 10, as mentioned in the Methods section. Also, we used our GPU-based gene expression quantification tool, Paean, which performed better in cross-validation with quantitative RT-PCR results. The results of Paean are consistent with other pipelines. We cited an updated version of Paean that included the comparison with other tools in analyzing AS for consistency. The manuscript on the new Paean version is being reviewed in another journal, and we included the PDF of that manuscript (Fig. 3 in the Paean manuscript) in the revised documents.

(6) Along the same lines, the finding that male age-related AS events are linked to Alzheimer's disease somewhat contradicts epidemiological studies that show that even after adjusting for age, women still have a greater risk of developing Alzheimer's than men. The authors show a significant overlap with AD GE events in females but don't explain the discrepancy.

We appreciate the editor’s comment regarding these discrepancies with the epidemiological studies. Previous studies suggested that the disease manifestations of Alzheimer’s Disease (AD) showed sex differences in AD phenotypes, including cognitive decline and brain atrophy [8]. The analyses on the sex/age effect of AD are indeed pretty complex, depending on the molecular criteria (GE or AS vs epidemiological data) in distinct studies, probably due to the difficulty in capturing how environmental exposures interact with biological pathways. We hope to bring up three related points regarding this concern, which were also discussed in the revised manuscript.

• As we have mentioned in the Discussion section, an early study investigated the relationship between age, sex, and cognitive function in a large cohort of 17,127 UK Biobank participants [9]. Their study highlighted more apparent age-related changes in cognitive function among men, suggesting a potential vulnerability of men to cognitive decline with age. Their main conclusion is consistent with our findings.

• While men and women can both suffer from Alzheimer's disease, women are more likely to be diagnosed, possibly due to longer lifespans and potential differences in brain structure or other factors. Although women exhibit a higher overall risk of AD, they may also have distinct molecular compensatory mechanisms that influence disease progression.

• To avoid the age effect, in our AD datasets, including ROSMAP, we filtered the samples over 90 years old to match the number of both sexes and the age distribution between the AD and control groups. Our analysis avoided the age biases in comparing AD and control, suggesting the crucial roles of sBASEs in AD during male aging.

Moreover, for gene expression (GE), we showed distinct patterns of AD-related genes in females with AS. These two molecular processes do not necessarily have the same functional impact. AS changes may precede or contribute to disease onset in different ways compared to GE alterations. Our study came up with the underlying mechanisms linking cognitive disorders and alternative splicing (AS) at a higher molecular resolution.

(7) Could the authors explain which sBASE subset they used for their random forest prediction model and what was the rationale?

We are sorry for missing the details in selecting sBASEs (sex-biased age-associated splicing events) for the random forest prediction model. We specifically used sBASEs that exhibited specific sex-biased changes in splicing associated with aging. This subset of sBASEs was chosen in terms of those that could also be detected in the ROSMAP AD dataset due to different sequencing depths or technical biases across datasets. These sBASEs were further input to a prediction model with the feature selection algorithm RFE, and then evaluated their contributions. In the revised manuscript, we added the details of this selection in the Methods (Page 7).

(8) The breakpoint analysis is particularly interesting. Can this be speculated to correlate with the recent non-linear multi-omic aging patterns observed by Shen et al in Nature Aging?

Thank you for highlighting the interesting aspects of our breakpoint analysis and suggesting its potential correlation with the non-linear aging patterns observed by Shen et al.

Shen et al. observed two prominent crests around the ages of 45 and 60 using omics data. Similarly, we also identified the non-linear aging patterns with two breakpoints in our analysis. However, there are some notable differences in specific breakpoints between these two studies, resulting from the breakpoint definition, as well as the sample preparations. According to the response in Author response image 2, the differences come from the following aspects:

The definition of breakpoints vs crests:

• Crests represent age-related molecular changes at each time point across the human lifespan. They indicate the number of molecules that are differentially expressed during aging (q < 0.05), without considering individual expression levels.

• Our breakpoints, in contrast, are identified after filtering the chronological trends based on the expression levels and calculating the rate of change at each age point using sliding windows. Breakpoints are defined as local maxima where the distance to the nearest minimum, relative to the global maximum, exceeds 10%. We indeed found some local wide peaks around 60 in some tissues, shown in Figure S10, however, we excluded these due to our strict cutoffs.

The sequenced biosamples:

• Whole-blood vs Peripheral Blood Mononuclear Cells (PBMC): As mentioned in previous responses, in GTEx, whole blood samples from donors were sequenced, whereas their study used PBMCs. Whole blood contains all blood components, including red blood cells, platelets, granulocytes (e.g., neutrophils), lymphocytes, and monocytes, while PBMCs only represent a subset of white blood cells, primarily consisting of lymphocytes (T cells, B cells, NK cells) and monocytes, excluding granulocytes and erythrocytes. Gene expression changes observed in whole blood capture the contributions from neutrophils and other granulocytes, which are absent in PBMC analyses (as shown in Figure S11C and Author response image 2). Additionally, whole blood can serve as a readily accessible biomarker source for testing age-related diseases without the need for cell separation, making it a more practical option for clinical applications.

• For both studies, we share a tissue, which is skin, we looked at the non-linear changes during aging and found the same two breakpoints: 43 and 58.

Sex-specific analysis in females and males:

• The main object of our analysis is to compare the differences in aging rates between sexes. Notably, the identified breakpoints may differ when sex effects are not taken into account, highlighting the importance of analyzing males and females separately.

We have added the following statements to further clarify this connection: ‘Our analysis observed the nonlinear aging patterns with two breakpoints, which is consistent with recent findings (Nature Aging, 2024), with differences in specific age points due to the sex differences as well as tissue diversities.’ (Page 14), and ‘These breakpoints could represent key junctures in the aging process that align with the non-linear patterns of aging and disease progression.’ (Page 15)

(9) Minor - the authors should refer to figures in the Discussion. They do so in some cases but this needs to be more extensive.

Thank you for pointing this out. In response, we have reviewed the Discussion section and added references to relevant figures where appropriate. In the section discussing the discrepancies between the profiles of GE vs. AS, we now refer to Figure 3 to highlight the earlier onset of different transcriptomic resolutions (Page 12); When describing the sex-specific age-associated AS changes and their associations with Alzheimer’s disease, we have added references to Figure 4 (Page 12); In the discussion of estrogen-mediated regulation of splicing factors, we have referred to Figure 5A, which detail the construction of RBP-RNA regulatory network integrating muti-dimensional data obtained through several orthogonal state-of-the-art approaches (Page 14).

Reference:

(1) Ferreira, P.G. et al. The effects of death and post-mortem cold ischemia on human tissue transcriptomes. Nature communications 9, 490 (2018).

(2) Wucher, V., Sodaei, R., Amador, R., Irimia, M. & Guigó, R. Day-night and seasonal variation of human gene expression across tissues. PLoS Biology 21, e3001986 (2023).

(3) Shen, X. et al. Nonlinear dynamics of multi-omics profiles during human aging. Nature aging, 116 (2024).

(4) Zhou, X. et al. Splicing factor SRSF1 promotes gliomagenesis via oncogenic splice-switching of MYO1B. The Journal of clinical investigation 129, 676-693 (2019).

(5) Soheili-Nezhad, S., Ibáñez-Solé, O., Izeta, A., Hoeijmakers, J.H. & Stoeger, T. Time is ticking faster for long genes in aging. Trends in Genetics 40, 299-312 (2024).

(6) Brouillette, M. Gene length could be a critical factor in the aging of the genome. Proceedings of the National Academy of Sciences 121, e2416630121 (2024).

(7) Keele, G.R. et al. Global and tissue-specific aging effects on murine proteomes. Cell reports 42(2023).

(8) Ferretti, M.T. et al. Sex differences in Alzheimer disease—the gateway to precision medicine. Nature Reviews Neurology 14, 457-469 (2018).

(9) Foo, H. et al. Age-and sex-related topological organization of human brain functional networks and their relationship to cognition. Frontiers in aging neuroscience 13, 758817 (2021).